# Crafting moiré superlattices in twisted complex oxide–transition metal dichalcogenide heterostructures

Rahul [1,12], Puneet Kaur [1,12], Jia-Yuan Sun[2,12], Jun-Ding Zheng[3,12], Shih-Chieh Lin[4,12], Yi-De Liou [1], Chia-Chun Wei[1], Shih-Chao Chang[1], Yu-Chen Liu[1], Ru-Long Gou [1], Ting-Hua Lu [5], Yann-Wen Lan [5], Tse-Ming Chen [1], Yi-Chun Chen [1], Yung-Chang Lin [6,7], Kazu Suenaga [7], Chun-Gang Duan [3] ✉, Wei-Ting Hsu [4,8,9] ✉, Chih-Wei Luo [2,8,10] ✉ & Jan-Chi Yang [1,11] ✉

Moiré superlattices, arising from overlaying atomic layers with slight mismatch or rotation, have transformed the study of emergent electronic and quantum phenomena beyond those of the constituent materials. Expanding this paradigm, here we demonstrate moiré superlattice formation at the interface between strongly correlated oxides and two-dimensional layered materials. The integration of complex oxides, a classic family of strongly correlated electron systems with transition metal dichalcogenides, enables the realization of an emerging class of moiré-engineered heterostructures that may potentially extend beyond conventional van der Waals systems. Herein, we reveal the presence of moiré superlattices in oxide-WS$_2$ heterostructures across varying twist angles and demonstrate highly tunable moiré periodicity as well as ultrafast charge transfer in these oxide-transition metal dichalcogenide systems. Direct observation of moiré exciton minibands confirms the emergence of moiré electronic structures, enabling twist-tunable discrete quantum states and unconventional charge dynamics. In combination with continuum modeling and density functional theory, our results elucidate the intricate interplay between moiré periodicity, quantum confinement, and band-flattening effects. By harnessing the synergy between complex oxides and layered materials, this work establishes a versatile platform for engineering artificial quantum states, providing previously inaccessible insights into correlated quantum phenomena and quantum material engineering.

In recent years, moiré superlattices/patterns in 2D materials have drawn considerable attention for their ability to host novel quantum phenomena with potential implications for next-generation electronic and quantum technologies. The moiré superlattices emerge either by introducing rotational misalignment, i.e., twist between the two layers, or by superimposing two dissimilar materials with lattice mismatch. The resultant long-wavelength interference pattern leads to a spatially modulated potential landscape. The moiré superlattice hosts a variety of emergent phenomena, such as tunable flat bands, which can host numerous strongly correlated phenomena. Building upon the concept

A full list of affiliations appears at the end of the paper. ✉e-mail: cgduan@clpm.ecnu.edu.cn; wthsu@phys.nthu.edu.tw; cwluoep@nycu.edu.tw; janchiyang@phys.ncku.edu.tw

of rotational misalignment, various moiré homobilayer systems have been extensively explored, including magic-angle twisted bilayer graphene (MATBG)[1,2], twisted monolayer–bilayer graphene[3], twisted double-bilayer graphene[4–6], and twisted transition metal dichalcogenides (TMDs)[7]. Furthermore, a variety of heterobilayer systems comprising 2D layered materials (2DLM), including TMDs, graphene, hexagonal boron nitride (hBN), etc.[8–10] have been investigated, leveraging inherent lattice mismatch to engineer moiré effects. In addition, theoretical studies have predicted a rich variety of moiré material platforms capable of hosting exotic correlated electronic states[11–13]. The moiré potential generated in these vertically stacked van der Waals (vdW) 2D layered structures profoundly alters the band structure, giving rise to a range of emergent quantum phenomena, including superconductivity, moiré excitons, moiré phonons, ferroelectricity, and magnetism. These phenomena collectively pave the way for next-generation quantum devices and correlated quantum materials[14].

Apart from traditional 2D materials, artificially created quasi-2D systems derived from strongly correlated 3D oxides have attracted significant interest[15]. The quasi-2D materials derived from strongly correlated complex oxides exhibit rich phase diagrams with a variety of emergent phenomena such as metal-insulator transitions, high-temperature superconductivity, polar vortices, and multiferroicity, owing to their intricate phase diagrams and strong electronic correlations[16–18]. Recent advances in freestanding (FS) thin-film fabrication now allow integration of such membranes with tailored properties into diverse material systems[19,20]. By incorporating an epitaxial sacrificial layer between the substrate and the functional oxide, highly crystalline, ultrathin, FS membranes that mimic 2D materials can be produced after selective etching[21]. Along this vein, moiré patterns have been demonstrated in twisted bilayer oxide nanomembranes[22]. The FS approach offers exceptional flexibility in materials choice, strain engineering, and precise control over orientation and twist angles[23]. Once decoupled from the substrate, FS quasi-2D complex oxide membranes can be seamlessly integrated with conventional vdW 2D materials, where the interplay between weakly and strongly correlated systems is anticipated to yield emergent quantum phenomena[24,25]. These exotic heterostructures may further expand the paradigm of moiré twistronics, as the intrinsic order parameters of complex oxides could strongly influence the electronic, magnetic, and optoelectronic properties of 2D materials. In a pioneering study, moiré patterns were observed in a CVD-grown, substrate-aligned MoS$_2$-STO system[26]. However, to the best of our knowledge, the realization of tunable moiré superlattices and their associated phenomena through deterministic twist-angle control in oxide–TMD heterostructures has yet to be achieved.

In this work, we demonstrate the formation of moiré patterns in an unconventional, symmetry-engineered twisted heterostructure composed of epitaxial complex oxide thin films and monolayer WS$_2$. This platform enables tunable moiré superlattices, symmetry breaking, and moiré excitons in twisted complex oxide–TMD systems, which have not been previously experimentally realized. The interlayer interactions at the twisted interface enable modulation of orbital hybridization, ultrafast charge transfer, and spin-lattice interactions. These effects collectively induce pronounced twist angle-dependent changes in the band structure and enable tunable control over spin injection and charge transfer dynamics on femtosecond to picosecond timescales. Unlike conventional epitaxial heterostructures, oxide–TMD twisted systems feature atomically sharp vdW-coupled interfaces that enable controlled tuning of electronic and quantum states, as further supported by continuum modeling and density functional theory (DFT). Furthermore, the demonstration of moiré patterns in complex oxide–2D TMD twisted heterostructures offers deeper insight into their interfacial physics and points toward potential pathways for their incorporation into future device platforms. By extending moiré twistronics beyond conventional 2D materials, this work expands the scope for leveraging the correlated electronic phenomena inherent in complex oxides, laying the foundation for an emerging class of hybrid quantum materials.

## Results

### Interface engineering and moiré superlattice formation in twisted oxide–TMD heterostructures

In order to derive the moiré superlattices at the interface of perovskite complex oxides and 2D TMDs, interface engineering plays a crucial role in integrating material systems with inherently distinct lattice symmetries. Figure 1(a-c) illustrates the typical crystal structure of the ABO$_3$ SrTiO$_3$ (STO) perovskite system in (001), (110), and (111) orientations. The (001) surface consists of alternating SrO and TiO$_2$ layers forming a square lattice, while the (110) surface presents a rectangular arrangement. Owing to this substantial crystallographic disparity, neither orientation supports the formation of periodic moiré patterns with WS$_2$ (Fig. S1). In contrast, the (111)-oriented STO surface exhibits a pseudo-hexagonal lattice, arising from repeating SrO$_3$ (AO$_3$) and Ti-cation (B-cation) layers stacked along the [111] direction. When viewed in-plane, this hexagonal network can be represented by a primitive orange parallelogram with an effective lattice parameter of $\sqrt{2}a_{STO}$ ($a_{STO} = 3.905$ Å), as shown in Fig. 1d. The WS$_2$ monolayer also forms a hexagonal lattice with parameter a$_{WS_2}$ (Fig. 1e). A nearly commensurate match is achieved when the effective lattice of STO corresponds to $\sqrt{3}a_{WS_2}$, as indicated by the orange dashed parallelogram (Fig. 1(d-e)), highlighting the compatibility of STO(111) with the TMD lattice. Furthermore, the moiré periodicity is highly sensitive to the relative orientation between the STO [1$\bar{1}$2] direction and the WS$_2$ [100] zigzag direction, leading to pronounced twist-angle-dependent variations (Fig. 1(f-h)). Complementary simulations (Fig. S2) confirm that moiré periodicity in WS$_2$/oxide heterostructures decreases with increasing twist angle, underscoring their ability to host tunable moiré superlattices as a platform for emergent quantum states.

### Visualization of tunable moiré periodicity in twisted WS$_2$-STO heterostructures

To realize moiré patterns between STO(111) and WS$_2$, the first step is to fabricate quasi-2D STO(111) thin films with thickness comparable to monolayer WS$_2$. Here, we adopted freestanding (FS) thin-film techniques to produce ~4 nm (~10 unit cells) FS-STO. STO thin films were grown on commercial STO(111) with La$_{0.7}$Sr$_{0.3}$MnO$_3$ (LSMO) as a sacrificial layer, which has been widely used to obtain ultrathin FS-STO membranes[23]. After selective acid etching, high-quality FS-STO membranes were released (Fig. S3), as detailed in the *Experimental Section*. XRD and AFM characterizations, shown in Figs. 2a-c and S4-S6, confirm epitaxial single-crystal quality for both the as-deposited STO/LSMO/STO and the released FS-STO. The FS-STO membranes were then transferred onto TEM grids or designated substrates, showcasing the versatility of the FS approach for integrating strongly correlated oxides into twisted bilayer (TBL) systems. The next step involved synthesizing monolayer WS$_2$ using conventional CVD with WO$_3$ and S precursors (Fig. S7)[27]. The CVD-grown flakes exhibit triangular morphology, as shown in Fig. 2d, with a thickness of ~0.65 nm consistent with single-layer 2H-WS$_2$[28,29], as further confirmed in Fig. 2e. Raman (Fig. S8) and XPS (Fig. S9) spectra verify the structural and chemical quality, while STEM imaging in Fig. 2f confirms W-terminated monolayers with zig-zag edge[30]. It is noteworthy that the triangular edges observed by OM correspond to the zig-zag crystallographic direction, providing a reference for alignment. Critically, when the WS$_2$ zig-zag edge is aligned with the [1$\bar{1}$2] direction of STO(111), the system yields the largest moiré periodicity, which we define as the 0° twist configuration. This baseline enables systematic tuning of moiré periodicity by varying the twist angle. Transferring WS$_2$ was achieved by selectively etching the SiO$_2$ layer on Si[31]. OM images (Fig. S10) acquired

before and after transfer demonstrate that the monolayer WS$_2$ retains its morphology, while PL measurements (Fig. S11) exhibit nearly unchanged emission intensities and peak positions. These monolayers were subsequently transferred onto FS-STO(111), forming oxide−TMD twisted heterostructures, with the in-plane and cross-sectional stacking schematically illustrated in Fig. 2g-h.

The formation of moiré patterns in these twisted oxide−TMD heterostructures was directly observed by STEM imaging of a WS$_2$/STO heterostructure transferred onto a TEM copper grid. In Fig. S12, WS$_2$ atop FS-STO(111) is resolved together with EDS mapping. Because both WS$_2$ and STO(111) possess hexagonal symmetry and compatible lattice constants, their stacking produces periodic interference, giving rise to distinct moiré lattices. The STEM images in Fig. 2i reveal a stark contrast between different regions: the freestanding oxide region (bottom right) shows no moiré patterns, whereas the oxide−TMD stacked region (top left) exhibits intricate, well-defined moiré lattices. This contrast underscores the unique structural characteristics of the oxide−TMD twisted heterostructure configuration, where lattice mismatch and twist angle act as critical tuning parameters that govern moiré periodicity and, in turn, may strongly correlate with interfacial electronic, optical, and quantum phenomena. Plane-view HRTEM images at twist angles of 3°, 5°, and 15° (Fig. 2j-l) reveal corresponding moiré wavelengths of 4.8, 3.5, and 1.2 nm. Cross-sectional HRTEM (Fig. S13) further confirms atomically sharp interfaces between STO and monolayer WS$_2$, establishing a robust platform for interfacial moiré engineering. However, the mere observation of moiré patterns does not necessarily signify strong interlayer interactions, and further investigations are required to elucidate the nature and extent of moiré-induced electronic modifications.

## Observation of moiré exciton minibands in twisted WS$_2$-STO heterostructures

To directly probe the impact of interlayer interactions and the emergence of moiré electronic structures, we measured differential reflectance (DR) and photoluminescence (PL) spectra of WS$_2$/STO twisted heterostructures at T = 5 K. As shown in Fig. 3a-c, DR and PL provide complementary access to excitonic transitions. For twist angles $\theta \to 0°$, two discrete resonances (mX$_1$ and mX$_2$) appear on the low-energy side of the monolayer WS$_2$ A exciton (X$_A$) (Fig. 3a, b). Apart from defect-bound excitons (X$_D$) and minor differences in relative intensity, the exciton features resolved by DR and PL are identical (see comparison in Fig. S14). As $\theta$ decreases from 8° to 0°, all excitonic resonances redshift, with the most pronounced evolution for $\theta \leq 2°$, consistent with the formation of moiré exciton minibands.

To establish the origin of these resonances, and to exclude contributions from defect emission and strain inhomogeneity, we combine temperature-, spatial-, power-, and polarization-resolved PL analyses (details in Figs. S14−16). We conclude that the mX$_1$ and mX$_2$ are K-valley neutral excitons arising from moiré minibands in WS$_2$/STO twisted heterostructures, whereas the low-energy X$_D$ (gray band in Fig. 3b) are defect-bound excitons: (i) As the temperature increases (Fig. 3d-e), the defect-bound X$_D$ quenches rapidly, whereas mX$_1$ and mX$_2$ persist to elevated temperatures with thermally broadened linewidths, behavior characteristic expected for interband optical transitions. (ii) Spatial PL maps show that mX$_1$, mX$_2$, and X$_A$ are uniform across the flakes (Fig. S15), in contrast to the strongly localized X$_D$; the absence of X$_D$ in DR spectra is also consistent with its weak oscillator strength. (iii) Power-dependent PL shows that the intensities of mX$_1$, mX$_2$, and X$_A$ scale linearly with laser power, as expected for neutral excitons (Fig. S14). (iv) Finally, polarization-resolved PL provides decisive valley-selective evidence. mX$_1$, mX$_2$, and X$_A$ exhibit robust valley polarization and coherence, firmly assigning them to K-valley neutral excitons in WS$_2$. Moreover, the PL linear-polarization axis co-rotates with the incident laser polarization, consistent with a rotationally symmetric moiré potential landscape[32,33] (Fig. S16).

Having established the K-valley neutral excitons, we fit the DR spectra with a Lorentz-oscillator model (Figs. 3f, S17) to quantify the twist-angle-dependent energy evolution. As the twist angle decreases,

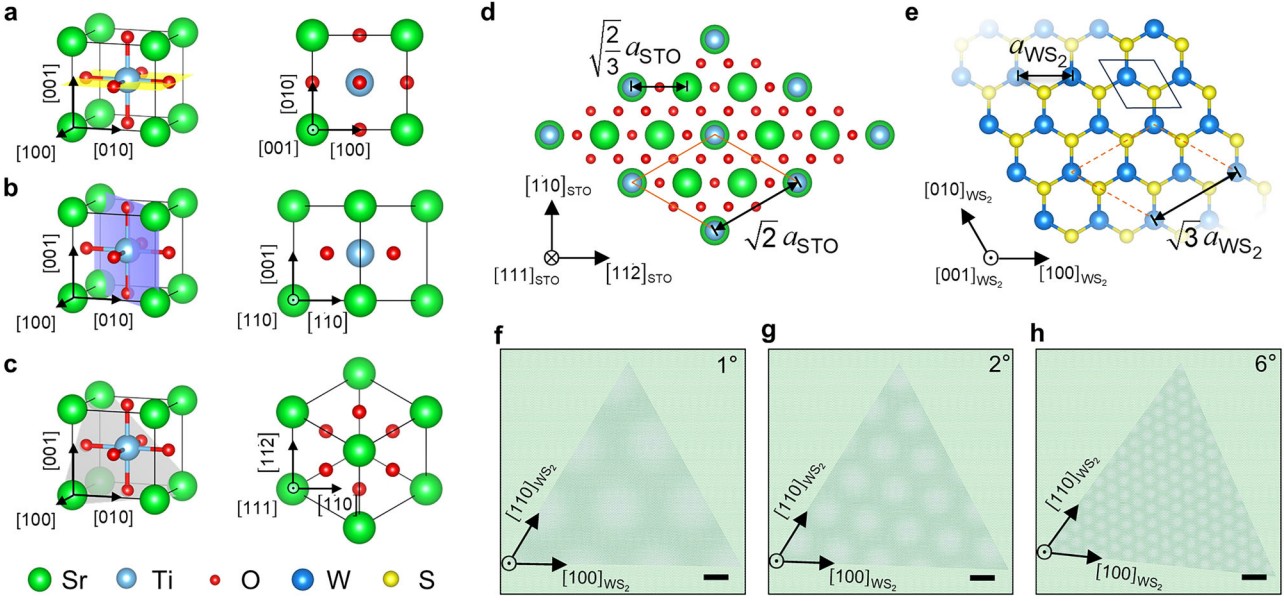

**Fig. 1 | Structural configurations and moiré pattern formation in perovskite oxide-TMD twisted heterostructure.** Schematic illustrations of the plane-view projection of STO in the **a** (001), **b** (110), and **c** (111) orientations, highlighting their distinct symmetry and atomic arrangements. **d** The (111)-oriented STO surface exhibits a hexagonal lattice, with its effective primitive cell outlined by the orange solid parallelogram. The corresponding lattice parameter is $\sqrt{2}a_{STO}$, where $a_{STO}$ = 3.905 Å is the cubic lattice constant of STO. **e** The WS$_2$ monolayer forms a hexagonal lattice, with its primitive cell indicated by the gray solid parallelogram.

For direct comparison, the orange parallelogram from (**d**) is overlaid onto the WS$_2$ lattice (orange dashed parallelogram), showing that its dimensions coincide with $\sqrt{3}a_{WS_2}$ ($a_{WS_2}$ = 3.15 Å) and thereby illustrating the lattice-matching condition between WS$_2$ and STO. **f**–**h** Schematic representation of the simulated moiré patterns of triangular WS$_2$ flakes on SrTiO$_3$(111) at various relative twist angles. As the twist angle increases, moiré periodicity systematically decreases. Sr and W atoms are denoted by green and blue spheres, respectively. (Scale bar = 5 nm).

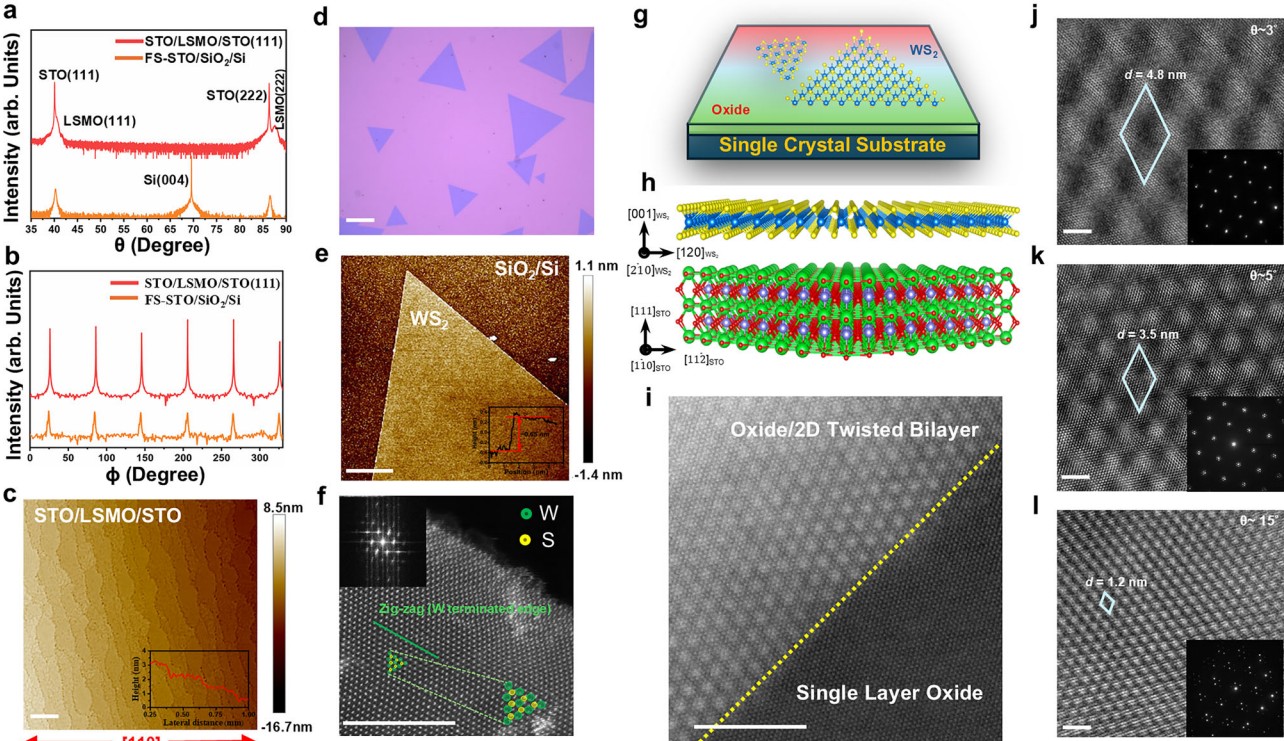

**Fig. 2 | Fabrication and structural characterization and the formation of perovskite oxide-TMD twisted heterostructure. a** X-ray diffraction L-scan and **b** phi-scan of as-grown STO/LSMO/STO(111) and freestanding STO(111)/SiO$_2$, confirming single-crystalline thin-film quality. **c** AFM image of as-grown STO/LSMO/STO(111), showing the smooth surface morphology (scale bar = 300 nm). **d** Optical microscopy image of a CVD-grown WS$_2$ monolayer on a SiO$_2$ substrate (scale bar = 20 μm). **e** AFM image of a WS$_2$ monolayer, with the inset indicating a uniform thickness of ~0.65 nm (scale bar = 4 μm). **f** STEM image of a WS$_2$ monolayer, highlighting W-terminated zigzag edges of the triangular flake (scale bar = 5 nm). The inset shows the FFT of the full image, indicating the crystallinity of the WS$_2$. **g** Schematic plane-view and **h** cross-sectional illustration of the stacking configuration of STO–WS$_2$ twisted bilayers. **i** STEM image of a STO/WS$_2$ twisted bilayer showing moiré superlattices formed in the oxide–TMD heterostructure (scale bar = 5 nm). **j–l** HRTEM images of twisted oxide–TMD heterostructures at twist angles of 3°, 5°, and 15°, together with the corresponding SAED patterns (insets), revealing the tunable rotational misalignment and moiré patterns between oxide and TMD layers (scale bar = 2 nm).

mX$_1$ and mX$_2$ shift to lower energies (Fig. 3g). The observed phenomena, (1) the emergence of discrete low-energy resonances (mX$_1$, mX$_2$) and (2) their redshift with decreasing θ, provide clear signatures of moiré exciton minibands. As illustrated in Fig. 3c, these moiré excitons can be understood as confined exciton states in periodic moiré potential. Reducing the twist angles (increasing the moiré periodicity) effectively broadens the potential width, thereby weakening the confinement effect in exciton states, which ultimately leads to the observed redshift of exciton energy[34–40].

**Continuum modeling and DFT of moiré potential in oxide–TMD heterostructures**

As established above, the low-temperature spectra reveal clear moiré exciton features. To further quantify the moiré potential depth, we calculated exciton minibands using a continuum effective Hamiltonian for the WS$_2$ exciton in a periodic potential, H = H$_0$ + Δ(r), where H$_0$ includes the kinetic and exchange terms and Δ(r) represents the lowest-harmonic moiré potential[40]. For each twist angle θ, we diagonalized the Hamiltonian in a plane-wave basis to obtain the miniband dispersions. The optical response was then simulated by evaluating dipole matrix elements (oscillator strengths) of the exciton Bloch states[40]. Full procedures and parameters are provided in the methods section. Figure 4a-b show representative minibands and calculated absorption spectra for θ = 1° and 6°. At small angles, the model yields multiple bright transitions and an overall energy redshift, consistent with the measurements. The calculated optical resonances shown in Fig. 4c exhibit a rapid redshift and the emergence of the mX$_2$

resonance when the twist angle is below ~2°, in remarkable agreement with our experimental observations. We subsequently varied the potential depth V$_m$, to reproduce the angle dependence of the mX$_1$ and mX$_2$ peak energies. This procedure yields a moiré potential depth of V$_m$ = 50 ± 10 meV (Fig. 3g), which quantitatively benchmarks the strength of the interfacial moiré potential in WS$_2$/STO twisted heterostructures.

To elucidate the origin of the moiré potential, we performed DFT calculations for WS$_2$/STO(111) heterostructures to study how local stacking configurations modulate the WS$_2$ band structures. Figure 4d displays a representative crystal structure for the high-symmetry AA stacking configuration, and additional atomic registries (AB and AC stacking configurations) are provided in Fig. S18. In order to facilitate the observation of the band structure of the WS$_2$ monolayer, in Fig. 4e we present the effective band structure by band unfolding. Firstly, we can notice that since the STO(111) surface is a polar surface, the Fermi level (E$_F$) of the heterostructures enters the original conduction band of STO. Secondly, the valence band of WS$_2$ enters the original band gap of STO, while the conduction band of WS$_2$ enters the conduction band of STO and is slightly higher than the E$_F$ of the heterostructure, resulting in a type-II band alignment. Finally, although the conduction bands of WS$_2$ and STO are energetically aligned, the K-point component of the WS$_2$ conduction band exhibits minimal mixing with that of STO. Meanwhile, the valence band of WS$_2$ lies within the band gap of STO and remains entirely non-hybridized. We then constructed a two-dimensional grid of relative in-plane displacements d, calculated the band structure at each point, and extracted the WS$_2$ K–K direct

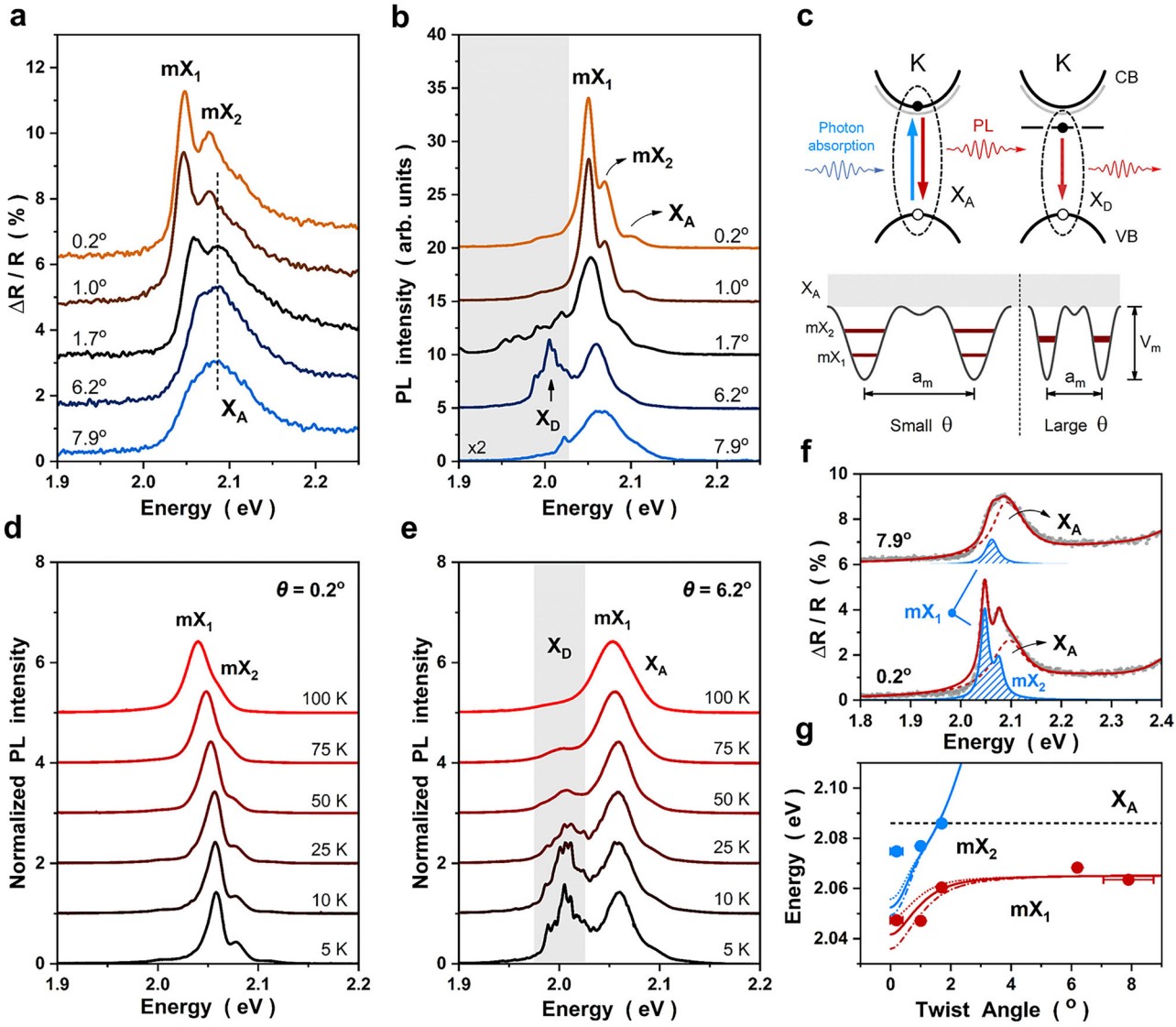

**Fig. 3 | Emergence of moiré exciton minibands in WS$_2$-STO twisted hetero-structures.** Low-temperature **a** DR and **b** PL spectra versus twist angle θ. As the twist angle approaches 0°, two moiré excitons (mX$_1$ and mX$_2$) emerge on the low-energy side of the WS$_2$ A exciton (X$_A$). In PL, defect-bound excitons (X$_D$) are observed at a lower energy (gray area). **c** Top: schematic band diagram illustrating that DR probes interband absorption, whereas PL probes emission; X$_D$ is absent in DR due to its weak oscillator strength. Bottom: schematic moiré potential with depth V$_m$ and period a$_m$. Reducing the twist angle increases the moiré periodicity, expanding the potential width and weakening the confinement effect, which leads to the observed redshift of moiré excitons. Temperature-dependent PL spectra of the θ = 0.2° (**d**) and θ = 6.2° (**e**) samples. At elevated temperatures, the X$_D$ peak quenches rapidly; mX$_1$ and mX$_2$ persist, albeit with thermally broadened linewidths. This supports an interband excitonic origin rather than defect recombination. **f** Comparison of fitted DR spectra (red solid curves) for small (0.2°) and large (7.9°) twist angles. The DR spectra contain mX$_1$/mX$_2$ states (blue-striped region) in addition to X$_A$ peak (red dashed curves). **g** Energy evolution of moiré excitons (mX$_1$ and mX$_2$). The black dashed line indicates the X$_A$ energy. Both mX$_1$ and mX$_2$ exhibit redshift as the twist angle decreases, with mX$_2$ appearing only below 2°. Also shown are continuous-model simulations that reproduce the observed twist-angle dependence, with the dotted, solid, and dashed-dotted curves corresponding to V$_m$ = 40, 50, and 60 meV, respectively. The twist angle is determined as the average of the three triangle edge azimuths relative to STO $[11\bar{2}]$, with error bars given by the standard deviation.

bandgap. The resulting modulation, $\Delta(d) \equiv \Delta E_c(d) - \Delta E_v(d)$, is shown in Fig. 4f. The black solid lines represent the supercell of the WS$_2$/STO(111) heterostructures, whereas the white dashed lines represent the primitive unit cell of WS$_2$. The x and y axes denote the in-plane displacements (relative translation) between the WS$_2$ and STO lattices. The resulting peak-to-peak modulation amplitude of bandgap is ~70 meV. Further Fourier transform of $\Delta(d)$ yields first-shell components of $V_0 = 5.07$ meV and $\psi = 58.23°$, in close agreement with the experimental data and continuum-model simulation ($V_0 = 5.55$ meV and $\psi = 60.0°$).

In twisted TMD bilayers, the origin of the moiré potential has been widely discussed. Commonly considered mechanisms

include interlayer electronic hybridization (typically weak at the K valleys) as well as strain-induced deformation and polarization fields[39–41]. While first-principles calculations based on hybridization alone often predict shallow moiré potentials, recent experimental work has instead implicated interfacial strain/reconstruction as a key contributor to deep moiré potentials exceeding 100 meV[34,42,43].

In the WS$_2$/STO(111) system, however, our DFT calculations, performed without explicitly including interfacial strain, already yield a bandgap modulation of ~70 meV, comparable in magnitude to the experimental observations. This comparison suggests that the dominant origin of the moiré potential at this 2D-oxide interface

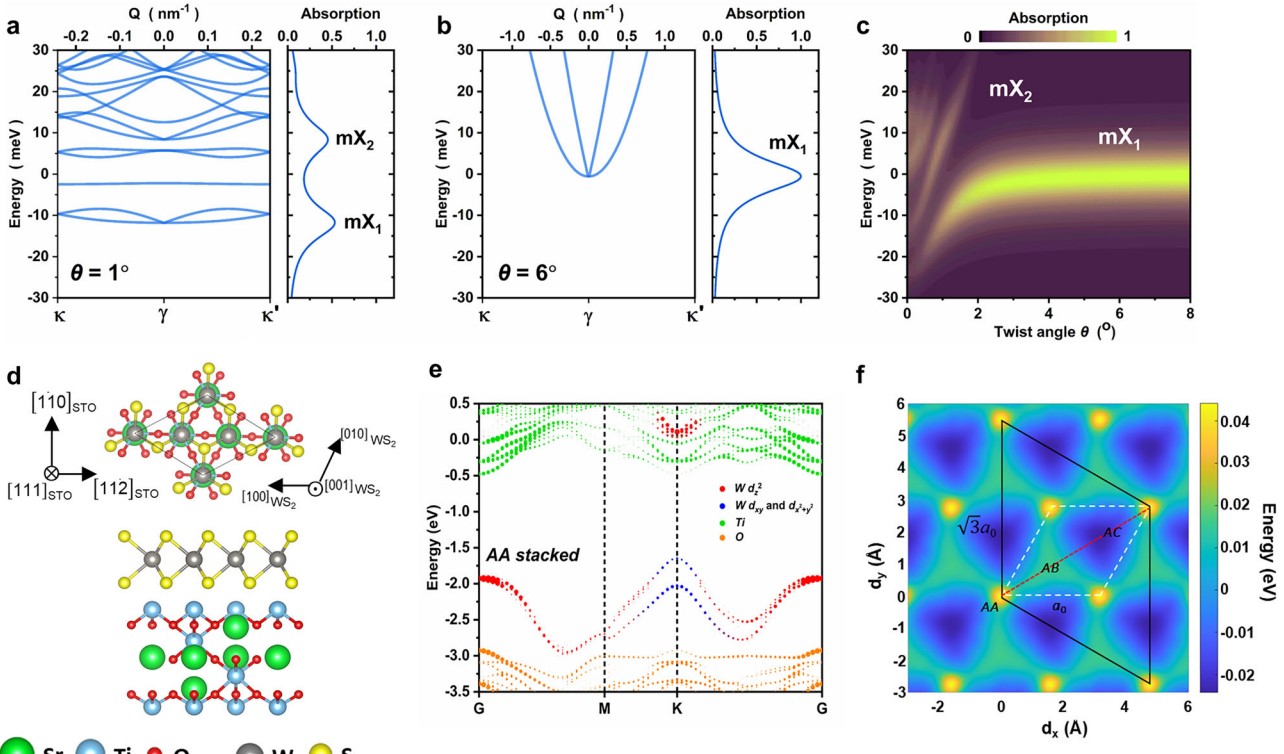

**Fig. 4 | Moiré-exciton model and DFT calculation. a, b** Moiré-exciton continuum-model miniband dispersions (left) and corresponding optical resonances (right) for twist angles θ = 1° and 6°. **c** Color-coded contour plot of moiré-exciton resonances versus twist angle θ and energy. These simulations are calculated with $V_m$ = 50 meV. **d, e** Structural and electronic properties of WS₂/STO(111) in the high-symmetry AA stacking configuration. **d** Atomic stacking arrangement, where the AA registry corresponds to a √3 × √3 WS₂ monolayer matched to Ti-terminal STO(111). **e** Corresponding unfolded band structure highlighting the electronic characteristics of the heterostructure. **f** DFT calculations of the bandgap modulation Δ(d), where d represents the relative in-plane displacement between WS₂ and STO. The black solid lines represent the supercell of the WS₂/STO heterostructure, whereas the white dashed lines represent the primitive unit cell of WS₂.

may differ from that in twisted TMD bilayers. We hypothesize that the large moiré potential does not primarily originate from interlayer hybridization but rather from a stacking-dependent local interfacial dipole from the Ti layer. As noted earlier, while a small degree of hybridization is present in the conduction band of WS₂, the valence band in the AA-stacked configuration shows virtually no hybridization. In Fig. S19, we further provide the effective band structure of AB and AC stacked configurations. Their K point of WS₂ conduction band also rarely contains STO components. Therefore, from the perspective of composition, we can rule out hybridization as the main source of the moiré potential in the WS₂/STO(111) twisted heterostructures. To further support our viewpoint, the shift of valence-band relative $\Delta E_v(d)$-$\Delta E_F(d)$ and the shift of conduction-band relative $\Delta E_c(d)$-$\Delta E_F(d)$ are shown in Fig. S20. The calculations show that the WS₂ valence band, situated entirely within the STO bandgap and exhibiting no hybridization with STO states, dominates the band modulation, thereby confirming that hybridization is not the primary source of the moiré potential. The large moiré potential can be attributed to the local dipole moment generated by polar STO(111) surface dipole. This dipole perturbs the onsite energies of the W d orbitals that dominate the K-valley band edges, $d_{xy}/d_{(x^2-y^2)}$ at the VBM and $d_{z^2}$ at the CBM, thereby modulating the band gap. DFT calculations show that, within the lateral-sliding registry space, the two high-symmetry configurations that yield the largest band-gap contrast are the AA and AC stackings. In the AA (AC) registry, a Ti atom in STO is aligned directly beneath a W atom (S atom) of the WS₂ lattice. This registry-dependent dipole produces a substantial modulation of the out-of-plane electrostatic field, and hence of the band gap, giving rise to the observed moiré potential.

## Twist-angle-modulated charge transfer and spin dynamics in twisted TMD-oxide heterostructures

Beyond the prototypical WS₂/STO system, our approach to constructing twisted oxide–TMD heterostructures provides a broadly applicable platform for probing moiré-mediated physics, while also enabling integration with strongly correlated oxides that host functionalities largely absent in conventional TMDs, such as magnetism and ferroelectricity. Here we focus on the magnetic perovskite LSMO as a representative case (Fig. 5), with further demonstrations involving LaCoO₃, PbZrO₃, and PbTiO₃ shown in Fig. S21. Taking advantage of its dielectric sensitivity, second harmonic generation (SHG) serves as a powerful probe for twisted 2D systems[44,45]. Power-dependent SHG confirms quadratic scaling (exponents ≈ 2) in WS₂ on different substrates (Fig. S22), consistent with nonlinear optical theory[46]. SHG mapping shown in Fig. 5a reveals twist-angle–dependent intensity variations, and analysis of ΔSHG versus twist angle shows a distinct 60° periodicity in WS₂ on (111)-oriented oxides, with the effect further enhanced when a conducting magnetic LSMO layer (~2 nm) is introduced (Figs. 5b and S23). Furthermore, polarization-dependent SHG (Fig. 5c) exhibits the characteristic isotropic sixfold $D_{3h}$ symmetry of WS₂, which remains unchanged with twist angle, thereby excluding anisotropic strain as the origin of the observed SHG modulation[46,47]. Unlike conventional TMD bilayers, where SHG arises from coherent superposition[48], in WS₂/LSMO/STO the LSMO layer is centrosymmetric and SHG-inactive (Fig. S24), and its contribution is negligible compared with the large nonlinear susceptibility of WS₂ ($d_{eff}$ = 0.77 nm/V)[47,49]. Although biaxial strain can also generate an isotropic sixfold SHG pattern[50], our Raman spectra show no measurable shift in the $E'$ peak for twist angles between 3° and 60° (Fig. S25), corresponding to a strain <0.03%[51]. Such a small strain would induce an

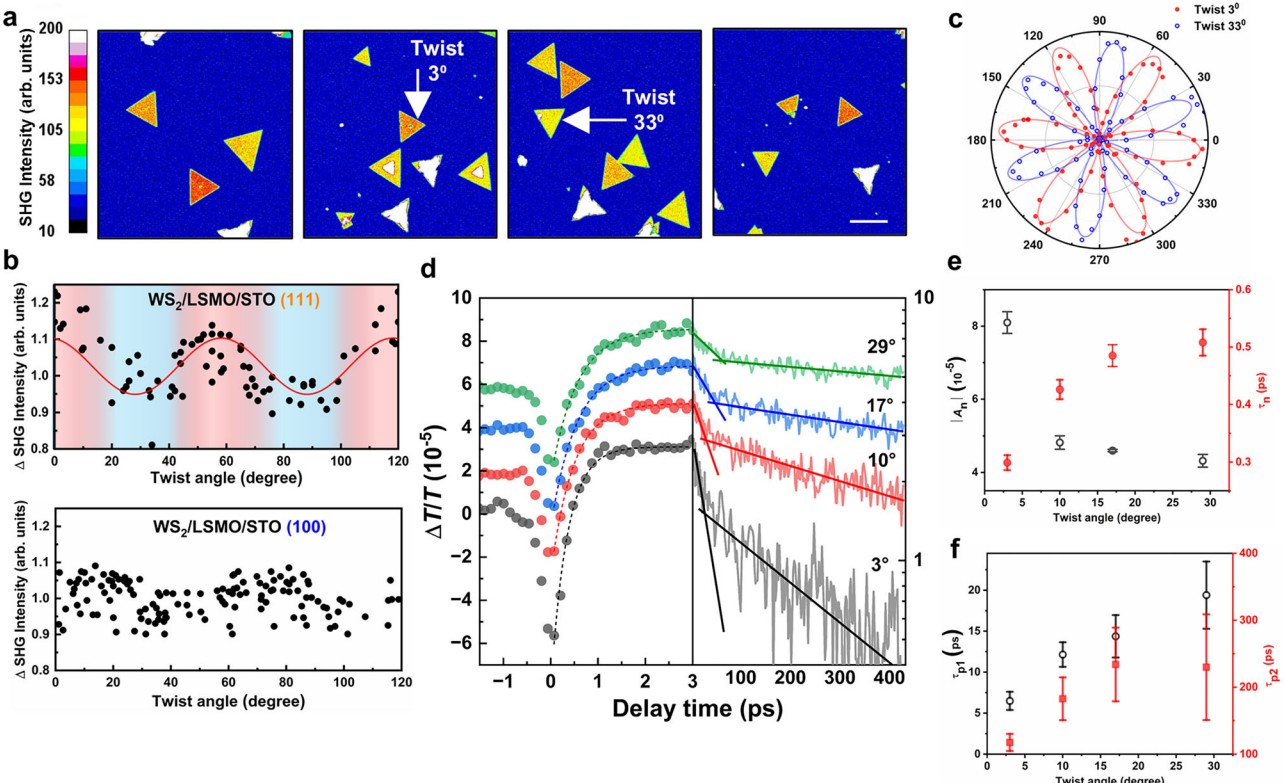

**Fig. 5 | SHG and pump-probe measurements of twisted WS₂-oxide heterostructures. a** SHG intensity maps of WS₂ on LSMO (111) substrates with different twist angles. The scale bar represents 30 μm. **b** Twist angle-dependent normalized SHG intensity changes (ΔSHG) for WS₂ on LSMO (111) substrates, including WS₂/LSMO (100), and WS₂/LSMO (111). Solid red lines represent the fitting of the sine function with a periodicity of 60⁰ to highlight the systematic variation. **c** Linear polarization dependence of the SHG intensity for WS₂ with twist angles of 3° and 33°, corresponding to (**a**). **d** Pump-probe transient transmittance (ΔT/T) signals of WS₂/LSMO/STO (111) at various twist angles. Dashed lines represent the

exponential fitting of the relaxation time $\tau_n$. Solid lines are the guide to the eyes. **e** Extracted amplitude $|A_n|$ and relaxation time $\tau_n$ of the negative $\Delta T/T$ component obtained from (**d**), as a function of twist angles. **f** Extracted fast relaxation time $\tau_{p1}$ and slow relaxation time $\tau_{p2}$ of positive $\Delta T/T$ component as a function of twist angles, revealing twist-dependent charge transfer and relaxation dynamics in the oxide-TMD heterostructures. Error bars represent ±1 standard error of the fitted parameters obtained from unweighted nonlinear least-squares fits to bi-exponential decay models with an offset $y_0$.

SHG modulation of <9.3%, which is significantly below the ~25% periodic modulation observed experimentally (Fig. 5b). These observations establish a direct link between twist angle and moiré superlattice formation, indicating that the SHG modulation primarily arises from twist-angle–dependent variations in the interlayer spacing, which evolve with stacking configuration and reach a minimum near 60°, thereby strongly influencing charge transfer[52–54]. Further PL spectra shown in Fig. S26 corroborates this mechanism, showing quenching at reduced interlayer spacing (large moiré periodicities), particularly at 0° and 60°[31,55]. Importantly, although the above heterostructures were prepared by wet-chemical transfer, SHG images and polar plots from dry-transferred samples with controlled twist angles (Figs. S27 and S28) show comparable features. Overall, these results indicate that the modulation of SHG intensity mainly originates from twist-controlled charge transfer, where variations in interlayer spacing regulate electron migration between oxides and WS₂, in turn modifying the dielectric response and SHG signals[56,57].

To correlate the magnetic interaction with charge transfer in twisted TMD/oxide heterostructures, ultrafast charge transfer dynamics were further studied using pump–probe spectroscopy[58–62]. Figure 5d shows transient transmittance (ΔT/T) spectra of WS₂/LSMO/STO(111) across various twist angles. The amplitude $|A_n|$ of the negative $\Delta T/T$ within the first ps increases as the twist angle decreases, with the strongest response observed peaking at 3°, consistent with enhanced charge transfer at smaller interlayer distances and larger moiré periodicities. Strong WS₂–LSMO coupling in these moiré states accelerates

charge transfer, reducing the relaxation time $\tau_n$ below 0.3 ps for small angles, comparable to TMD heterobilayers[63]. At longer delays, $\Delta T/T$ exhibits two decay channels: $\tau_{p1}$ in the few-ps range (orbital–lattice recovery) and $\tau_{p2}$ on the hundreds-ps scale (spin–lattice relaxation of LSMO). Both show strong twist-angle dependence: smaller twist angles yield shorter times, reflecting more efficient electron transfer. Figure 5e-f further highlights this systematic reduction, demonstrating direct control of interfacial coupling by twist angle. Because only majority-spin $e_g$ states of LSMO lie at the Fermi level, transferred electrons populate this channel, transiently strengthening double exchange between $Mn^{3+}$ and $Mn^{4+}$. This enhances ferromagnetic metallicity until orbital–lattice recovery re-localizes carriers[63,64]. The dependence of $\tau_{p1}$ and $\tau_{p2}$ on twist angle thus evidences spin-polarized charge injection and its coupling to lattice and spin relaxation. By tuning twist angle and combining TMDs with magnetic oxides, one can modulate charge-transfer efficiency and spin–lattice interactions, enabling control of spin injection and magnetization dynamics on femto- to picosecond timescales. Notably, the intrinsic hundreds-ps spin–lattice recovery sets a natural limit for magneto-optical switching, underscoring opportunities for engineering TMD/oxide-based spintronic devices.

## Discussion

Through the combination of freestanding oxide membranes and interfacial engineering, we have demonstrated the formation of moiré superlattices in unconventional oxide–TMD twisted

heterostructures. Using $WS_2$/STO(111) as a model system, our work establishes a versatile framework for crafting moiré lattices beyond conventional van der Waals bilayers. Structural imaging directly confirms the presence of moiré patterns, while low-temperature optical spectroscopy reveals two discrete, twist-tunable resonances ($mX_1$ and $mX_2$) that persist to elevated temperatures and exhibit clear signatures of moiré exciton minibands. Complementary continuum modeling reproduces the twist-angle evolution of these resonances and yields a moiré potential depth of $50 \pm 10$ meV, whereas density functional theory maps the band-edge modulation of ~70 meV. The approaches converge to identify the polar STO(111) interfacial dipole, rather than strong orbital hybridization, as the dominant origin of the moiré potential. These results provide a rigorous theoretical and experimental foundation for understanding exciton confinement and interfacial coupling in oxide–TMD moiré systems.

Looking ahead, our results establish a clear demonstration of moiré formation in $WS_2$/STO and $WS_2$/LSMO twisted heterostructures and position oxide–TMD interfaces as a versatile platform for exploring emergent quantum phenomena. The tunable moiré potential in these systems enables controlled modulation of exciton dynamics, charge transfer, and many-body interactions, where both interfacial dielectric screening and local strain fields arising naturally within the moiré supercell can act as complementary tuning knobs. Beyond $WS_2$, the framework we introduce can be broadly generalized to other TMDs with distinct excitonic and electronic structures, as well as to a wide family of functional oxides, including ferroelectrics, correlated nickelates, and magnetic manganites (Figs. S21 and S29). Coupling excitonic TMD layers with these oxide order parameters opens the door to moiré-engineered ferroelectric modulation, magnetically tunable valley polarization, interfacial Mott transitions, and spin-textured moiré minibands. Such tunability is particularly appealing for designing valleytronic elements, excitonic modulators, and quantum photonic components. At the same time, the intrinsic electrostatic, mechanical, and ferroic tunability of oxide membranes provides dynamic control of twist-dependent coupling that is not accessible in conventional van der Waals moiré systems. Altogether, this work not only demonstrates the feasibility of creating robust oxide–TMD moiré superlattices but also highlights their unique potential as reconfigurable building blocks for low-power optoelectronic switches, integrated quantum photonic elements, and multifunctional hybrid quantum technologies fully compatible with existing oxide electronics.

## Methods

### Preparation of freestanding $SrTiO_3$

STO thin films were fabricated by pulsed laser deposition (KrF, 248 nm) on $(La,Sr)MnO_3$ (LSMO)-buffered (111)-oriented STO substrates. The STO was deposited at an oxygen pressure of 100 mTorr at 700 °C with a laser power of 250 mJ and laser repetition rate of 10 Hz, while the LSMO was deposited at an oxygen pressure of 150 mTorr at 715 °C. The LSMO buffer was subsequently etched in HCl to release the freestanding STO films, which were transferred onto lacey carbon grids.

### Growth of monolayer $WS_2$

Monolayer $WS_2$ was synthesized by ambient-pressure Chemical Vapor Deposition (CVD) on $Si/SiO_2$ substrates. The substrates were ultrasonically cleaned in acetone, isopropanol, and water, rinsed with DI water, and dried at 60 °C. $WO_3$ (99.9%, Sigma-Aldrich) and S (99.9%, Sigma-Aldrich) served as precursors, placed at the center and inlet of the quartz tube, respectively. Growth was carried out at 900 °C with S maintained at 210 °C, using $Ar/H_2$ (10:1, 180 sccm) as carrier gas. After growth, the furnace was naturally cooled to room temperature.

### Wet transfer process

For TEM observations, the as-grown $WS_2$ flakes were transferred via a wet-transfer process from the $Si/SiO_2$ substrate onto lacey carbon grids containing freestanding STO(111) membranes. Poly(methyl methacrylate) (PMMA) was spin-coated (4000 rpm, 60 s) on the top of $WS_2$ samples and baked (120 °C, 1 min) to peel off $WS_2$ flakes. Afterward, the assembly was dipped into HF solution to etch the PMMA/$WS_2$ stack, which was then subsequently rinsed repeatedly with DI water to get rid of the contaminants from the etching. The PMMA/$WS_2$ stacked assembly was transferred onto a lacey carbon grid, and PMMA was removed with acetone and IPA. The resulting STO/$WS_2$ twisted bilayer heterostructures were further annealed (180 °C, 2 hr.).

### Structural characterization

L-scans and azimuthal φ-scans of STO and LSMO were carried out on a high-resolution synchrotron 8-circle X-ray diffractometer at the National Synchrotron Radiation Research Center (NSRRC), Taiwan. Measurements were performed at beamlines TLS-17B, TLS-13A1, and TPS-09A using a 10 keV monochromatic beam from a Si(111) double-crystal monochromator. The incident beam was shaped to $500 \times 1000$ μm with upstream slits, while downstream slits were employed to suppress background scattering.

Moiré patterns in $WS_2$/STO heterostructures at varying twist angles were examined by plane-view TEM and EDS using a JEOL JEM-2100F (Cs-corrected STEM at Core Facility Center, NCKU, Taiwan). Cross-sectional TEM samples were prepared by FIB milling and lift-out techniques. Monolayer $WS_2$ imaging was performed on Triple-C#1 (JEOL 2100 F, 60 kV, cold FEG, dodecapole correctors) with a probe current of ~15 pA, convergence semi-angle of 35 mrad, and inner acquisition semi-angle of 79 mrad. In the presented STEM images, higher electron intensity (brighter contrast) typically reflects regions with a higher atomic number (Z-contrast), originating from heavier atomic species and/or increased local specimen thickness, while lower intensity indicates lighter elements or thinner regions. In the FFTs calculated from the STEM images, brighter spots correspond to dominant spatial frequencies associated with periodic atomic arrangements, providing reciprocal-space information analogous to diffraction and enabling the identification of lattice periodicities. In the HRTEM images, contrast arises primarily from mass-thickness and diffraction effects, where darker regions generally correspond to thicker areas or regions with stronger electron scattering. In the EDS elemental maps, color brightness represents the relative elemental concentration, with higher brightness indicating higher local abundance of the corresponding element. In SAED, brighter spots indicate stronger diffraction intensity from certain lattice planes, primarily determined by the structure factor and crystal orientation.

### Surface analysis

The AFM measurements were carried out using Bruker's closed-loop commercial scanning probe microscope system equipped with Nanoscope 9.14 in tapping mode. The core-level XPS measurements (PHI VersaProbe 4) were performed using a scanning Al Kα x-ray source having an energy resolution of $\leq 0.5$ eV. All measured binding energies were referenced further w.r.t. the carbon 1 s signal.

### Optical characterization

Raman and PL spectra of monolayer $WS_2$ were collected using a confocal microscope with 532 nm continuous wave laser excitation equipped with a 100× (NA 0.9) objective, and an 1800 lines/mm grating under ambient conditions.

Low-temperature DR spectroscopy was performed using a home-built confocal microscope with a tungsten–halogen lamp as the light source. To precisely determine the twist angle, the $WS_2$ flakes were transferred onto (111)-oriented STO single crystal substrates with well-defined in-plane [$1\bar{1}2$] and [$1\bar{1}0$] crystallographic directions. Reflected

signals were analyzed by a spectrometer with a liquid-nitrogen-cooled CCD. Samples were cooled to 5 K in a closed-cycle cryostat with three-axis nanopositioners, and illumination/collection was achieved via an objective lens. Here, the Nb-doped STO (0.05 wt%) substrate is used to facilitate the precise determination of the twist angle. The DR signal is defined as the reflectance difference between the $WS_2$ monolayer and the underlying STO, primarily influenced by multilayer interference and interband optical absorption of excitons. Details of the Lorentz oscillator fitting are provided in Supplementary Information S17.

SHG and Pump-probe measurements were performed using a Ti:Sapphire laser (800 nm, 200 fs, 5.1 MHz) integrated with a home-made laser-scanning confocal microscope (LSCM) at room temperature. A 20× objective (NA 0.45) was used for focusing excitation and collecting the signal in the backward direction for SHG measurements. Furthermore, the laser served as the excitation source, and polarization-dependent signals were recorded while rotating the samples in 5° steps over 360°. For pump-probe spectroscopy, samples were pumped at 400 nm (3.1 eV) and probed at 800 nm (1.55 eV) with fluences of 500 and 50 mJ/cm², respectively, to investigate ultrafast dynamics.

### Continuum effective Hamiltonian for moiré excitons

Moiré minibands were obtained by diagonalizing a continuum effective Hamiltonian for the center-of-mass (COM) motion of the lowest bright excitons in monolayer $WS_2$, including electron-hole exchange. The moiré potential is represented by a lowest-harmonic expansion, and the Hamiltonian is solved in a plane-wave basis[40]. Acting on the bright-exciton valley pseudospin doublet, the model reads: $H = H_0 + \Delta(\boldsymbol{r})\tau_0$, where $H_0 = (E_0 + \hbar^2\boldsymbol{Q}^2/2M)\tau_0 + J|\boldsymbol{Q}|\tau_0 + J|\boldsymbol{Q}|[\cos(2\phi_Q)\tau_x + \sin(2\phi_Q)\tau_y]$. Here, $\boldsymbol{Q}$ is the exciton momentum with orientation angle $\phi_Q$, M is the $WS_2$ exciton COM effective mass, $E_0$ is the bright-exciton energy at Q = 0, and $\tau_0$, $\tau_{x,y}$ are the identity and Pauli matrices in valley space. The moiré potential is expanded as: $\Delta(\boldsymbol{r}) \approx \sum_{j=1}^{6} V_j e^{i\boldsymbol{b}_j\cdot\boldsymbol{r}}$, $\boldsymbol{b}_j = \boldsymbol{G}_j - \boldsymbol{G}_j'$ with $\boldsymbol{G}_j$ ($\boldsymbol{G}_j'$) the reciprocal lattice vectors of the top (bottom) layer. Requiring a real-valued $\Delta(\boldsymbol{r})$ and $C_3$ rotational symmetry gives $V_{1,3,5} = V_0 e^{i\psi}$ and $V_{2,4,6} = V_0 e^{-i\psi}$. We denote the potential depth (peak-to-valley modulation) by $V_m$. Unless otherwise noted, simulations use $M = 0.73m_0$[65], $\psi = 60°$, and $J = 0.4\text{eV}\cdot\text{Å}$, where $m_0$ is the free-electron rest mass. The potential depth $V_m$ is determined by reproducing the measured evolution of the $mX_1$ feature as a function of twist angle θ.

### DFT calculations

We carry out first-principles calculations by Vienna Ab initio Simulation Package (VASP)[66,67] with the Projector Augmented Wave (PAW) method, and apply the Generalized Gradient Approximation (GGA) and the Perdew-Burke-Ernzerhof (PBE) exchange-correlation functional[68,69]. The plane-wave energy cutoff is set to 500 eV, and Γ-centered $8 \times 8 \times 1$ k points grid is adopted for Brillouin zone sampling. The self-consistent and force convergence are $1.0 \times 10^{-6}$ eV and 1 meV/Å, respectively. The distance between the adjacent layers is kept at 20 Å to eliminate interlayer interaction, and the van der Waals interaction is corrected by the DFT-D3 method[70]. The full relaxed structure files used in the work are provided in Supplementary Data 1.

## Data availability

All data that support the findings of this study are presented in the Manuscript and Supplementary Information or are available from the corresponding author upon request. Source data are provided with this paper.

## Code availability

The code that was used to analyze the experimental data is available from the corresponding authors upon request.

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

## Acknowledgements

This work was supported by the National Science and Technology Council (NSTC), Taiwan, under grant numbers NSTC 112-2112-M-006-020-MY3, 112-2112-M-007-036-MY3, 113-2119-M-A49-001-MBK, 113-2112-M-A49-020-MY3, 113-2119-M-A49-001-MBK, 113-2112-M-A49-020-MY3, 113-2124-M-006-010, 114-2811-M-006-054, 114-2811-M-006-041, and 114-2124-M-006-003. W.-T.H. acknowledges the support from the Yushan Fellow Program from the Ministry of Education of Taiwan (Grant MOE-109-YSFMS-0002-001-P1). J.-D.Z acknowledges the support from the National Science Foundation for Young Scientists of China (Grant No. 12504105). The authors sincerely thank MSSORPS Co., Ltd. for their high-quality TEM sample preparation and preliminary analysis. We also extend our gratitude to Ms. Chi-Ren Luo for FIB sample preparation (EM025200, FEI, Helios NanoLab G3 CX) and Ms. Shih-Wen Tseng for

Cs-STEM observations (EM000800, JEOL JEM-2100F) conducted at the Core Facility Center, National Cheng Kung University (NCKU), Taiwan. Additionally, we thank Mr. You-Tong Lu, Mr. Chi-Yuan Chang, and Mr. Li-Shu Wang for their assistance with preliminary AFM measurements, and Prof. Yen-Lin Huang and Dr. Le Thi Quynh for their support in the initial XRD analysis. We also sincerely thank Prof. Zhen Chen for his valuable guidance in the TEM analysis, as well as Prof. Tay-Rong Chang and Dr. Christophe De Beule for their insightful theoretical suggestions and discussions. K.S. and Y.C.L. acknowledge JSPS-KAKENHI (21H05235, 22F22358, 22H05478), the JST-CREST program (JPMJCR20B1, JMJCR20B5), ERC "MORE-TEM" (951215), and the JSPS A3 Foresight Program. This research was also partially supported by the Higher Education Sprout Project, Ministry of Education, Taiwan, through the Headquarters of University Advancement at National Cheng Kung University (NCKU).

## Author contributions

Rahul and P.K. processed the growth of FS oxide thin films, optimized monolayer TMD growth and fabricated twisted oxide-TMD-based heterostructures, interpreted/analyzed data, and co-wrote the manuscript draft. J.-Y.S. carried out SHG and pump-probe measurements under the supervision of C.-W.L. and co-wrote the manuscript draft. J.-D.Z. and C.-G.D. performed DFT calculations and analyzed theoretical results. S.-C.L. conducted the optical measurements and analyzed the data under the guidance of W.-T.H. S.-C.L., J.-D.Z., and W.-T.H. interpreted the PL and continuum modeling results. C.-C.W. and Y.-C.L. (Yung-Chang Lin) processed TEM measurements, Y.-D.L., S.-C.C., Y.-C.L. (Yu-Chen Liu), and K.S. contributed to structural characterization and analysis. R.-L.G. carried out the dry transfer procedure under the supervision of T.-M.C. T.-H.L., Y.-W.L., Y.-C.C., K.S. and T.-M.C. provided valuable interpretation of the data. All authors actively participated in discussions. J.-C.Y. led the project, conceived the main idea, and, along with Rahul and P.K., designed the study. Rahul, P.K., J.-Y. S., J.-D.Z., and S.-C.L. contributed equally to this work.

## Competing interests

The authors declare no competing interests.

## Additional information

[1]Department of Physics, National Cheng Kung University, Tainan, Taiwan. [2]Department of Electrophysics, National Yang Ming Chiao Tung University, Hsinchu, Taiwan. [3]Key Laboratory of Polar Materials and Devices (MOE) and Department of Electronics, East China Normal University, Shanghai, China. [4]Department of Physics, National Tsing Hua University, Hsinchu, Taiwan. [5]Department of Physics, National Taiwan Normal University, Taipei, Taiwan. [6]National Institute of Advanced Industrial Science and Technology (AIST), Tsukuba, Japan. [7]The Institute of Scientific and Industrial Research (SANKEN), The University of Osaka, Osaka, Japan. [8]National Synchrotron Radiation Research Center, Hsinchu, Taiwan. [9]Research Center for Applied Sciences, Academia Sinica, Taipei, Taiwan. [10]Institute of Physics, National Yang Ming Chiao Tung University, Hsinchu, Taiwan. [11]Center for Quantum Frontiers of Research & Technology (QFort), National Cheng Kung University, Tainan, Taiwan. [12]These authors contributed equally: Rahul, Puneet Kaur, Jia-Yuan Sun, Jun-Ding Zheng, Shih-Chieh Lin. ✉e-mail: cgduan@clpm.ecnu.edu.cn; wthsu@phys.nthu.edu.tw; cwluoep@nycu.edu.tw; janchiyang@phys.ncku.edu.tw

