## [Transparent Peer Review file · Nature Communications]

Crafting Moiré Superlattices in Twisted Complex Oxide–Transition Metal Dichalcogenide Heterostructures

Corresponding Author: Professor Jan-Chi Yang

Version 0:

Reviewer comments:

Reviewer #1

(Remarks to the Author)

Reviewer #2

(Remarks to the Author)

Reviewer #3

(Remarks to the Author)

The authors investigate the effects of artificially engineered moiré superlattices formed by stacking and twisting a vdW-WS₂ on complex oxide materials (STO(111) and LSMO(111)), including both a freestanding oxide and thin films grown on substrates. The impact of varying twist angles is explored primarily through electron microscopy and optical characterizations. The topic is timely and potentially impactful, as hybrid heterostructures combining 2D materials with complex oxides could lead to intriguing emergent phenomena.

However, the presentation of the work lacks clarity, coherence, and focus. For instance, the study begins with WS₂/freestanding STO and then shifts to WS₂/LSMO thin films on STO substrates without clearly articulating the rationale or connection between these systems. As a result, the work contents are disjointed. Future readers are taken from one experimental setup to another without a well-defined conceptual link or progression.

In addition, the scientific depth is limited (e.g., orbital hybridization and charge transfer mechanisms at the interfaces are only superficially addressed). The methodology lacks twist angle control, which is a key factor in current moiré engineering, and the overall approach does not demonstrate significant innovation beyond standard characterization techniques. It appears that the authors simply identified various twisted regions after spreading WS₂ flakes onto oxide layers, rather than systematically engineering specific twist angles. The experimental findings also do not show substantial novelty compared to twisted 2D/2D (e.g., WSe₂/WSe₂) systems, aside from the change in material class. As such, the conceptual advance and impact of the work are not clearly established.

Given these concerns, particularly the lack of methodological rigor, insufficient experimental clarification, and limited scientific insight, I do not find the manuscript suitable for publication in Nat. Commun.

Below are my comments:

1. What is the role of complex oxide materials here? While the abstract and introduction suggest that electronic interactions at the WS₂/oxide interfaces are central to the work, the manuscript lacks a detailed investigation into these interfacial effects. In particular, there is no in-depth analysis of atomic-level interactions around the interface or electronic reconstruction (e.g., orbital hybridization, charge transfer), which are expected when coupling weakly correlated vdW materials with strongly correlated complex oxides. To support the claims and deepen the physical understanding, the authors could perform DFT calculations to examine band structure evolution and flat band formation as a function of twist angle (see this example, Nat.

Commun. 12, 5601 (2021)) for relevant methodology. This would also clarify whether the oxide layers contribute to moiré-induced flat bands.

Given that one of the key motivations stated in the abstract and introduction is to explore the interplay between vdW and correlated oxide systems, I strongly recommend that the authors include a more rigorous theoretical or spectroscopic investigation to resolve interfacial orbital interactions and emergent phenomena, which are not well understood in the presented work. Also, reference sample measurements (e.g., WS₂ on standard inert substrates, LSMO, and STO) are missing experimentally to identify the observed phenomena.

2. In lines 124–126, the authors state that the moiré pattern becomes smeared as the thickness of the bottom layers increases. However, they do not provide a physical explanation, references, or experimental support for this statement. Additional discussion is needed to clarify the underlying mechanism (whether it relates to diminished interfacial coupling, increased surface roughness, strain relaxation, or other effects in thicker films). Then, the authors should explicitly explain how this observation motivated their use of ultrathin oxide layers in the study.

3. It is unclear how precisely the twist angles presented in Fig. 3 were controlled or determined. For clarity, the authors should include representative images or spectral maps corresponding to the different twist angles that give rise to varied optical responses.

4. This manuscript would benefit from including atomic force microscopy (AFM) and X-ray diffraction (XRD) 2 θ scans of both the freestanding single-crystalline oxide layers on Si and the grown thin films on STO in Si as they can support for the claimed abrupt interface engineering.

5. Regarding the emergence of two discrete excitonic quantum states (Fig.3), I am curious whether the authors have also tested with the case of WS₂/LSMO. Also, have the authors investigated the temperature dependence of these excitonic states? Temperature-dependent measurements could help confirm their quantum-confined nature and distinguish them from other possible spectral features such as defect-bound excitons or inhomogeneous broadening effects.

6. What is the thickness of STO and LSMO layers used in this work? Is there any thickness dependence? I see the PLD growth conditions used in this work for both films are rather unconventional (e.g., 10 Hz). I am curious whether all the grown films were epitaxial.

7. In Fig. 5, the authors describe a comparison with “various complex oxides,” but only two oxides (STO and LSMO) are included, while SiO₂ is not a complex oxide. Therefore, the term “various” seems overstated in this context. Furthermore, all presented systems exhibit SHG signals, but it is unclear whether the authors performed control experiments on bare STO and SiO₂ substrates without WS₂. Both substrates are known to potentially exhibit non-trivial SHG responses due to surface or defects or local symmetry effects, so attributing SHG solely to moiré superlattices requires stronger clarification. To support their interpretation, the authors need to include angle-dependent SHG measurements of the bare substrates (STO and SiO₂). This would help confirm whether the SHG signals observed in WS₂/oxide stacks indeed originate from moiré-induced effects rather than intrinsic substrate contributions.

8. Are there any established strategies to precisely control twist angles between two-dimensional (2D) materials or FS membranes? If so, have the authors applied one of the methods in this work?

Version 1:

Reviewer comments:

Reviewer #1

(Remarks to the Author)

The authors have provided a comprehensive and well-structured rebuttal that clearly reflects a major revision effort. I appreciate the significant amount of new experimental and theoretical work that has been now included into the revised manuscript. The inclusion of low-temperature PL studies, power and temperature dependencies, spatial uniformity analysis, and polarization-resolved measurements provides a much more complete and convincing picture of the excitonic features in these oxide-TMD heterostructures. The addition of continuum modeling and DFT calculations gives quantitative support to the moiré potential estimation and strengthens the overall interpretation. These new results, together with the expanded discussion on the role of complex oxides and improved methodological clarity (particularly regarding twist-angle control), considerably enhance both the scientific depth and coherence of the work.

Overall, I find that the authors have satisfactorily addressed most of the critical points raised in the initial review. The revised manuscript now presents a compelling and well-substantiated case for the formation of moiré excitons at oxide-TMD interfaces and demonstrates the broader potential of this hybrid materials platform.

That said, I am not fully satisfied with the response to my comment concerning the SHG intensity modulation (previous comment 5). In their reply, the authors discuss uniaxial strain and use the six-lobe symmetry of the SHG pattern to exclude strain effects. However, my original concern referred to isotropic biaxial strain, which can modulate SHG intensity without altering the symmetry of the polar pattern. This effect has been quantified, for example, in Xing, H., Liu, J., Zhao, Z. et al. "Quantifying the in-plane strain influence on second harmonic generation of molybdenum disulfide" Commun Phys 7, 382

(2024), (see Fig. 3c). Given this, it remains unclear why the authors attribute the SHG intensity modulation solely to charge-transfer effects and do not consider the possibility that isotropic biaxial strain -possibly arising from isotropic buckling within the moiré supercell- could play a role.

Aside from this remaining ambiguity regarding the SHG interpretation, I find the revision convincing and the manuscript substantially improved. The new data provide deeper insight and significantly strengthen the claims, and the study now reaches the level of rigor and completeness expected for publication in Nature Communications once this last issue is properly clarified.

Reviewer #2

(Remarks to the Author)

Reviewer #3

(Remarks to the Author)

I noticed that the authors have made great efforts and have satisfactorily addressed all of my previous comments. The revised manuscript has been significantly improved, particularly in the organization and logical connection of the content, as well as in the strengthened theoretical discussion on the moiré potentials and electronic band-gap modulations as a function of twisting angle.

These revisions provide a comprehensive understanding of the twist-angle-dependent atomic modulations at the surfaces of complex oxides and the resulting emergent physical phenomena, such as moiré excitons and charge-transfer effects. I also think this work will be of broad interest to both the oxide and 2D-materials research communities. Therefore, I am pleased to recommend publication after addressing the following comments below.

Comments:

1. In the abstract (and also at the end of the Introduction), the authors should include another emergent phenomenon (the twist-angle-dependent symmetry breaking and highly tunable ultrafast charge-transfer effect arising in the magnetic oxide-WS₂ system).
2. Please indicate the thickness of the LSMO films used in this work in both the main text and the Experimental section.
3. Please clarify why the thickness of the freestanding STO membrane was chosen to be 4 nm rather than a thicker one.
4. Please verify whether Fig. S5c is correctly presented. The peak positions and assignments appear inconsistent with those in Figs. S5a,b.
5. Have you measured the SHG response of WS₂ flakes on freestanding STO(111) membrane? Such data could further support the proposed symmetry breaking (polarization) effects at the STO surface and the associated (or enhanced) SHG signals.
6. What is the rationale for choosing WS₂ in this study? How do other 2D-TMDs behave when interfaced with complex oxides? Please discuss this in the main text.
7. It would be beneficial to briefly mention potential applications to help attract a broader readership.

Reviewer #4

(Remarks to the Author)

This manuscript study the artificially engineered moiré superlattices formed by stacking and twisting a vdW-WS₂ on complex oxide materials (STO(111) and LSMO(111)). The authors observe the moiré exciton inbands, which confirms the emergence of moiré electronic structures. The authors also provide DFT calculations to support their findings. After getting through the previous reviewer's comments and the report, I am delighted to recommend this work for publication in Nature Communications. The reason is that the authors have supported their observations by theoretical input, which is convincing.

Version 2:

Reviewer comments:

Reviewer #1

(Remarks to the Author)

I appreciate the response and additional experiments provided by the authors. Please be careful of the Raman notations - for

TMD monolayers the correct in-plane mode representation should be E' according to the monolayer symmetry. I believe this work meets the criteria for publication in Nature Communications.

Reviewer #2

(Remarks to the Author)

Summary of the revisions:

1. Detailed PL studies (energy evolution, power dependence, spatial uniformity vs. localization, and valley polarization/coherence) have been included to rule out defect- or strain-induced inhomogeneous broadening, thereby confirming the observed features as genuine moiré neutral excitons.
2. A new Figure 4, integrating both continuum modeling and DFT results, has been incorporated into the revised manuscript.
3. Calculations based on a standard continuum model have been performed, yielding a moiré potential depth of $\sim 50 \pm 10$ meV, consistent with the experimental observation of moiré minibands (detailed in new Figure 4a-c).
4. As suggested by the referees, DFT calculations have been incorporated in the new Figure 4d-f, showing that the moiré potential in $\text{WS}_2/\text{STO}(111)$ arises from registry-dependent dipolar fields of the polar $\text{STO}(111)$ surface, not interlayer hybridization, yielding ~ 70 meV bandgap modulation consistent with our experiment.
5. The supplementary materials now extend the STO/WS_2 case to other oxide/2D interfaces, showing that moiré phenomena can also arise more broadly and highlighting new opportunities for future studies.
6. Additional XRD, AFM, phi-scan, L-scan, and SHG measurements have been incorporated in the revised manuscript as suggested by the reviewers.
7. Detailed SHG measurements, including a comparison between strained and unstrained WS_2 , have been incorporated to rule out strain-induced contributions, as raised by Reviewer 1 and 2.
8. Inspired by the insightful comments of Reviewers 1 and 2, we have incorporated a focused discussion that links WS_2/STO moiré formation to concrete future research directions, which is now integrated into the revised Discussion.
8. In response to Reviewer 3's comment, a PCL-based dry transfer approach has been presented to demonstrate precise control over twist angles, with SHG measurements included as supporting evidence.
9. The revised manuscript now incorporates and discusses the new findings in main text (Figure 3 and Figure 4) as well as in the supplementary information (Figure S2-7, S11, S14-16, S18-24, S26-27)

General Responses:

We would like to begin by sincerely thanking all three reviewers for their constructive, positive, and insightful comments. Their feedback not only highlighted important points of clarification but also gave us the opportunity to carefully revise and substantially strengthen our work. The revisions now provide a more rigorous and coherent narrative, integrating both additional experimental evidence and theoretical modeling to address the key concerns raised.

Before responding to each comment in detail, we would like to give two general responses to all reviewers. From the reviewers' comments, we understand that there are two main concerns. The first concern is about the verification of the moiré excitons observed in the DR spectra and how to distinguish them from inhomogeneous broadening effects due to defects or strain. The second concern is about the absence of an estimation of the moiré potential depth derived from the experimental results.

To address the first concern, we rely on detailed photoluminescence (PL) studies. By examining the energy evolution, linear power dependence, spatial uniformity versus localization, and excitonic valley polarization/coherence, we exclude contributions from defects and strain inhomogeneity, supporting the conclusion that the observed features are indeed moiré neutral excitons. For the second concern, based on the observed spectral evolution, a standard continuum-model simulation gives a moiré potential depth of ~ 50 meV. Independent DFT calculations yield a value of 70 meV. The consistency among spectroscopy, continuum modeling, and first-principles theory supports the formation of the moiré miniband. The detailed discussions are provided below.

1. Identification of moiré-induced exciton features

As pointed out by the reviewer, inhomogeneous broadening effects are important issues that need to be carefully addressed. In addition to defect-bound excitons, we recognize that strain is a crucial factor in TMD moiré heterostructures. In rotationally aligned heterobilayers (e.g., MoSe₂/WSe₂), strain and small twist fluctuations can strongly modulate the local moiré periodicity and can induce atomic-scale stacking reconstruction, yielding mesoscopic domains [1-7]. All of these can have a great impact on excitonic states. In the revised manuscript, we have expanded and improved the discussion of these critical issues. In the following sections, we exclude the effects of defects and strain inhomogeneity by comparing the relevant spectral characteristics, and demonstrate that the observed features correspond to moiré excitons.

Figures R1(a-b) compares the PL and DR spectra of the 0.2° and 7.9° samples. We observe that all exciton features, including mX₁, mX₂, and X_A, appear in both spectra. Apart from some variation in the relative intensities in the PL spectra, where lower-energy excitons are relatively stronger, the main excitonic peaks remain consistent between PL and DR measurements. Figures R1(c-d) show the evolution of DR and PL spectra as the twist angle decreases from 8° to 0°. Both mX₁ and mX₂ exhibit a redshift with decreasing angle, further demonstrating the consistency between PL and DR results. Figure R1(e) presents power-dependent PL measurements, showing that the relative intensities of mX₁, mX₂, and X_A remain nearly

unchanged with increasing laser power. Figure R1(f) shows the integrated PL intensity as a function of power, showing linear power dependence for mX_1 , mX_2 , and X_A .

Figure R1. Moiré excitons in the $\text{WS}_2/\text{SrTiO}_3$ heterostructure. The PL (black) and DR (red) spectra measured at $T=5\text{K}$ for the (a) $\theta=0.2^\circ$ and (b) $\theta=7.9^\circ$ samples. (c) The DR spectra and (d) PL spectra as a function of twist angle θ . Note that defect-bound excitons (X_D) are observed in the low-energy region of the PL spectra (gray area). (e) Normalized power-dependent PL spectra of the $\theta=0.2^\circ$ sample, showing identical exciton features at different laser powers. (f) Integrated PL intensity as a function of laser power, showing linear power dependence for mX_1 (black squares), mX_2 (red circles), and X_A (blue triangles). The black line is a reference for linear power dependence.

To evaluate the thermal stability and distinguish between moiré excitons and defect-bound excitons, temperature-dependent PL measurements were conducted, as shown in Figures R2(a-b). The defect-bound exciton X_D quenches rapidly with increasing temperature, consistent with thermal escape of carriers from defect states. In contrast, the mX_1 and mX_2 peaks persist at elevated temperatures, albeit with thermally broadened linewidths. This thermal robustness supports that mX_1 and mX_2 arise from interband optical transitions, rather than from defect-related recombination. To further assess spatial uniformity and exclude extrinsic effects such as strain inhomogeneity or defects, spatially resolved PL mapping was performed. Figures R2(c-h) show optical images of the 0.2° and 6.2° samples, along with PL intensity maps of the main excitonic features, including mX_1 , mX_2 , and X_A . The excitonic peaks are uniformly distributed across the flakes, indicating that the underlying moiré potential is spatially homogeneous and not significantly influenced by strain inhomogeneity. In contrast, the X_D peak (highlighted in Figure R2h) exhibits strong spatial localization, a characteristic of excitons trapped at defect sites. Together, these observations confirm that mX_1 and mX_2 originate from moiré exciton states rather than from strain inhomogeneity or defects.

Figure R2. Uniform moiré excitons in the WS₂/SrTiO₃ heterostructure. Temperature-dependent PL spectra of the (a) $\theta=0.2^\circ$ and (b) $\theta=6.2^\circ$ samples. At elevated temperatures, the X_D peak quenches rapidly; mX₁ and mX₂ persist, albeit with thermally broadened linewidths. Optical images of the (c) $\theta=0.2^\circ$ and (f) $\theta=6.2^\circ$ samples. Spatially resolved PL maps of the (d-e) $\theta=0.2^\circ$ and (g-h) $\theta=6.2^\circ$ samples. The PL maps reveal a uniform distribution of mX₁, mX₂, and X_A across the flakes, while the defect-bound exciton X_D exhibits clear spatial localization.

Figures R3(a-b) show the polarization-resolved PL of the 0° sample. The main excitonic features, including mX₁, mX₂, and X_A, all exhibit a certain degree of circular and linear polarization (lower panels). The circular and linear PL polarizations arise from the valley polarization and valley coherence of K-valley excitons in WS₂[8]. The finite valley coherence observed for mX₁, mX₂, and X_A demonstrates that they are K-valley neutral excitons [8]. To exclude the effects of strain inhomogeneity, we rotated the linear polarization angle of the incident laser and found that the main axis of the PL polarization rotates accordingly, as shown in Figures R3(c-f). These results demonstrate that the observed linear PL polarization indeed originates from exciton valley coherence.

Figure R3. Polarization-resolved PL of the WS₂/STO heterostructure. (a) Circular- and (b) linear-polarization-resolved PL spectra for the $\theta=0.2^\circ$ sample. Upper panels: polarization-resolved PL; lower panels: degree of circular (ρ_C) and linear (ρ_L) PL polarization, defined as $((I_{\sigma^+} - I_{\sigma^-}) / (I_{\sigma^+} + I_{\sigma^-}))$ and $((I_H - I_V) / (I_H + I_V))$, respectively. (c-e) Polar plots of linear-polarization-resolved PL, showing that all excitons (mX₁, mX₂, and X_A) follow the laser polarization axis. The mX₂ and X_A intensities are scaled by 2 for clarity. (f) Analysis of the PL polarization axis,

showing consistent alignment with the laser polarization.

We note that several important studies have demonstrated the presence of structural/atomic reconstruction in TMD moiré systems [1-7]. Critically, these microscopically reconstructed domains can significantly affect the electronic and excitonic states, and may even give rise to 1D or 0D moiré structures. Although a detailed characterization of such features is beyond the scope of the present study, our linear-polarization-resolved PL measurements tentatively exclude the formation of 1D moiré structures in our sample. In a typical 1D moiré stripe phase, one would expect highly linear PL polarization with a fixed polarization axis that cannot be rotated by the incident laser. In this case, our results are more consistent with 2D moiré excitons, which exhibit valley coherence and rotate with the incident laser polarization. These results also motivate future surface-sensitive measurements, such as PFM and STM, to directly probe the moiré pattern, which could be a valuable direction for further study. Once again, we thank the reviewers for their insightful comments that led us to a deeper understanding of moiré exciton physics in this study.

2. Simulation and estimation of moiré potential depth

In the second part, based on the observed spectral evolution, by using a standard continuum-model simulation we estimated a moiré potential depth of ~ 50 meV. Independent DFT calculations yield a value of 70 meV. The agreement among spectroscopy, continuum modeling, and first-principles theory supports the formation of the moiré miniband. The detailed discussions are provided below.

As established above, the low-temperature spectra reveal clear moiré exciton features. To further quantify the moiré potential depth, we calculated exciton minibands using a continuum effective Hamiltonian for the WS₂ exciton in a periodic potential [9]. For each twist angle θ , we diagonalized the Hamiltonian in a plane-wave basis to obtain the miniband dispersions. The optical response was then simulated by evaluating dipole matrix elements (oscillator strengths) of the exciton Bloch states [9]. Moiré minibands were obtained by diagonalizing a continuum effective Hamiltonian for the center-of-mass (COM) motion of the lowest bright excitons in monolayer WS₂, including electron-hole exchange. The moiré potential is represented by a lowest-harmonic expansion and the Hamiltonian is solved in a plane-wave basis [9]. Acting on the bright-exciton valley pseudospin doublet, the model reads: $H = H_0 + \Delta(\mathbf{r})\tau_0$, where $H_0 = (E_0 + \hbar^2 \mathbf{Q}^2 / 2M)\tau_0 + J|\mathbf{Q}|\tau_0 + J|\mathbf{Q}|\left[\cos(2\phi_Q)\tau_x + \sin(2\phi_Q)\tau_y\right]$. Here, \mathbf{Q} is the exciton momentum with orientation angle ϕ_Q , M is the WS₂ exciton COM effective mass, E_0 is the bright-exciton energy at $\mathbf{Q}=0$, and $\tau_0, \tau_{(x,y)}$ are the identity and Pauli matrices in valley space. The moiré potential is expanded as: $\Delta(\mathbf{r}) \approx \sum_{j=1}^6 V_j e^{i\mathbf{b}_j \cdot \mathbf{r}}$, $\mathbf{b}_j = \mathbf{G}_j - \mathbf{G}'_j$ with \mathbf{G}_j (\mathbf{G}'_j) the reciprocal lattice vectors of the top (bottom) layer. Requiring a real-valued $\Delta(\mathbf{r})$ and C3 rotational symmetry gives $V_{1,3,5} = V_0 e^{i\psi}$ and $V_{2,4,6} = V_0 e^{-i\psi}$. We denote the potential depth (peak-to-valley modulation) by V_m . Unless otherwise noted, simulations use $M=0.73m_0$ [10], $\psi=60^\circ$, and $J=0.4$ eV·Å, where m_0 is the free-electron rest mass. The potential depth V_m is determined by reproducing the measured evolution of the mX₁ feature as a function of twist angle θ .

Figures R4b-c show representative minibands and calculated absorption spectra for $\theta=1^\circ$ and 6° . At small angles, the model yields multiple bright transitions and an overall energy redshift,

consistent with the measurements. The calculated optical resonances (Figure R4d) exhibit a rapid redshift and the emergence of the mX_2 resonance when the twist angle is below $\sim 2^\circ$, also in agreement with experiment. We subsequently varied the potential depth V_m to reproduce the angle dependence of the mX_1 and mX_2 peak energies. This procedure yields a moiré-potential depth of $V_m=50\pm 10$ meV (see Figure R4a).

In addition to the continuum-model analysis, we performed density functional theory calculations to further clarify the microscopic origin of the moiré potential in $WS_2/STO(111)$, as shown in Figures R4e-g. The calculations reveal a bandgap modulation of ~ 70 meV across different stacking registries, in close agreement with the ~ 50 meV potential depth obtained from both spectroscopy and continuum-model simulations. Importantly, the band structures (Figure 4f) show that while a small degree of hybridization is present in the conduction band, the valence band of WS_2 remains fully decoupled from the STO states, thereby excluding interlayer electronic hybridization as the dominant source of the moiré potential. Instead, the substantial modulation originates from the local electrostatic dipole field of the polar $STO(111)$ surface (Figure 4f), which perturbs the onsite energies of the W d orbitals at the K-valley band edges. This registry-dependent dipole mechanism accounts for the large moiré potential observed experimentally and distinguishes oxide-TMD interfaces from conventional twisted TMD bilayers, where hybridization alone fails to explain the experimental potential depths. Thus, the consistency between experiment, continuum modeling, and DFT calculations provides robust evidence for the emergence of moiré minibands in this hybrid platform. We sincerely thank the referees for raising these insightful points, which motivated us to provide this comprehensive clarification and thereby strengthen the manuscript. For further details and the full discussion, please refer to the revised manuscript and Supplementary Materials.

Figure R4. Estimation of moiré potential depth from continuum modeling and DFT calculations. (a) Energy evolution of moiré excitons (mX_1 and mX_2). The dashed line indicates the X_A energy. Both mX_1 and mX_2 exhibit redshift as the twist angle decreases, with mX_2

appearing only below 2° . (b-c) Continuum-model calculations of moiré-exciton miniband dispersions (left) and corresponding optical resonances (right) for twist angles $\theta = 1^\circ$ and 6° . (d) Evolution of moiré-exciton resonances as a function of twist angle θ . (e-f) Structural and electronic features of WS₂/STO(111) in the high-symmetry AA stacking configuration: (e) atomic stacking arrangement, where the AA registry corresponds to a $\sqrt{3} \times \text{WS}_2$ monolayer matched to Ti-terminated STO(111); (f) unfolded band structure showing the electronic characteristics of the heterostructure. (g) DFT-calculated bandgap modulation $\Delta(d)$, where d denotes relative in-plane displacement between WS₂ and STO. Black solid lines mark the supercell of the WS₂/STO heterostructure, while white dashed lines indicate the primitive unit cell of WS₂.

References

1. Weston, A., Zou, Y., Enaldiev, V. *et al.* Atomic reconstruction in twisted bilayers of transition metal dichalcogenides. *Nat. Nanotechnol.* **15**, 592–597 (2020).
2. Rosenberger, Matthew R., *et al.* Twist Angle-Dependent Atomic Reconstruction and Moiré Patterns in Transition Metal Dichalcogenide Heterostructures. *ACS Nano* **14**, 4550–4558 (2020).
3. Bai, Y., Zhou, L., Wang, J. *et al.* Excitons in strain-induced one-dimensional moiré potentials at transition metal dichalcogenide heterojunctions. *Nat. Mater.* **19**, 1068–1073 (2020).
4. Shabani, S., Halbertal, D., Wu, W. *et al.* Deep moiré potentials in twisted transition metal dichalcogenide bilayers. *Nat. Phys.* **17**, 720–725 (2021).
5. Li, H., Li, S., Naik, M.H. *et al.* Imaging moiré flat bands in three-dimensional reconstructed WSe₂/WS₂ superlattices. *Nat. Mater.* **20**, 945–950 (2021).
6. Weston, A., Castanon, E.G., Enaldiev, V. *et al.* Interfacial ferroelectricity in marginally twisted 2D semiconductors. *Nat. Nanotechnol.* **17**, 390–395 (2022).
7. Zhao, S., Li, Z., Huang, X. *et al.* Excitons in mesoscopically reconstructed moiré heterostructures. *Nat. Nanotechnol.* **18**, 572–579 (2023).
8. Jones, A., Yu, H., Ghimire, N. *et al.* Optical generation of excitonic valley coherence in monolayer WSe₂. *Nature Nanotech* **8**, 634–638 (2013).
9. MacDonald, A.H. *et al.* Topological Exciton Bands in Moiré Heterojunctions. *Phys. Rev. Lett.* **118**, 147401 (2017).
10. Kim, K.W. *et al.* Intrinsic transport properties of electrons and holes in monolayer transition-metal dichalcogenides. *Phys. Rev. B* **90**, 045422 (2014).

Point-to-point responses: Reviewer #1 & 2

Reviewer Comment:

In this manuscript, the authors report the fabrication and characterization of twisted heterostructures composed of freestanding SrTiO₃ (111) thin membranes and monolayer WS₂. Using a combination of electron microscopy, optical spectroscopy (differential reflectance, PL and SHG), and ultrafast pump-probe measurements, the authors claim to observe moiré superlattice formation and associated quantum phenomena, including moiré exciton minibands, twist-angle dependent charge transfer, and nonlinear intensity modulation. This study aims to expand the landscape of moiré systems by integrating complex oxides known for strong correlations and tunable order parameters with van der Waals materials. The central finding that moiré patterns form between monolayer WS₂ and STO (111) is clearly supported by high-resolution TEM and diffraction data. The fabrication method involving freestanding epitaxial oxide membranes is technically solid and indeed offers a promising new route for interfacial engineering.

However, the manuscript suffers from a number of significant weaknesses that make me conclude it does not meet the standards of *Nature Communications*. While the experimental work is promising, particularly the demonstration of moiré pattern formation in hybrid oxide-TMD heterostructures, the interpretation of the data requires significant revision. The speculative claims need to be supported by theoretical calculations, and additional optical measurements (especially low-temperature PL) are needed to substantiate key conclusions.

Response:

We sincerely thank the referee for their thoughtful and constructive comments. Their feedback highlighted several important shortcomings in the original submission and provided valuable directions that have substantially improved both the scope and depth of our work. In response, we carefully revised the manuscript with major additions to address these concerns. Specifically, we now provide comprehensive theoretical support through continuum modeling and DFT calculations, as well as expanded optical spectroscopy (including low-temperature PL), directly substantiating our claims of moiré exciton minibands, interfacial charge transfer, and twist-dependent quantum phenomena. These additions place our interpretations on a solid quantitative foundation and clarify the microscopic mechanisms at the oxide-TMD interface. The specific changes and clarifications are detailed point by point in the following sections.

Reviewer Comment:

1. The interpretation of spectroscopic data relies heavily on speculative claims of *quantum confinement, orbital hybridization, moiré exciton minibands, and band flattening*. No theoretical calculations or simulations are provided to support these claims. Without even basic electronic band structure modeling or a discussion of excitonic selection rules in this hybrid system, the interpretation goes well beyond what is warranted by the data. These parts of the manuscript should be either significantly revised or supported by theoretical input. For instance, claiming exciton trapping within a moiré potential is highly speculative in the absence of any estimation of the potential depth and periodicity relative to the exciton Bohr radius.

Response:

We sincerely thank the referee for raising this important concern. Their comment prompted a major revision of our manuscript, leading us to incorporate rigorous theoretical continuum modeling and first-principles calculations to directly support our spectroscopic interpretations.

First, to quantify the moiré potential depth, we calculated exciton minibands within a continuum effective Hamiltonian for the WS₂ exciton in a periodic potential, $H=H_0+\Delta(r)$, where H_0 collects the kinetic-exchange terms and $\Delta(r)$ is the lowest-harmonic moiré potential [1]. For each twist angle θ , we diagonalized the Hamiltonian in a plane-wave basis to obtain the miniband dispersions. The optical response was then simulated by evaluating dipole matrix elements (oscillator strengths) of the exciton Bloch states [1]. Revised Figures 4a-b show representative minibands and calculated absorption spectra for $\theta=1^\circ$ and 6° . At small angles, the model yields multiple bright transitions and an overall energy redshift, consistent with the experimental observations. The calculated absorption spectra (Figure R5) exhibit a rapid spectral redshift and the emergence of the mX₂ resonance when the twist angle is below $\sim 2^\circ$, in agreement with the experimental observations. We subsequently varied the potential depth V_m to reproduce the angle dependence of the mX₁ and mX₂ peak energies. This procedure yields a moiré-potential depth of $V_m=50\pm 10$ meV (see Figure R5). At small twist angles ($\theta < 2^\circ$), the moiré period in WS₂/STO exceeds ~ 5 nm, well above the ~ 1 -2 nm Bohr radius of the K-valley A exciton. In this regime, the exciton center-of-mass experiences a slowly varying periodic potential, so pronounced moiré effects and exciton minibands are expected. These results provide direct theoretical grounding for the assignment of mX₁ and mX₂ as moiré exciton minibands. For detailed descriptions of the continuum modeling, including the framework, parameter selection, and comprehensive analysis, please refer to the revised Results section, Figures 4a–c, and Methods, where the methodology and outcomes are presented in full.

Second, we performed DFT calculations of WS₂/STO(111) across different stacking registries (AA/AB/AC). These reveal a type-II band alignment, with the WS₂ valence band lying entirely inside the STO gap and non-hybridized, while the K point of conduction band shows only minimal overlap with STO states. Mapping the registry-dependent bandgap yields a modulation amplitude of ~ 70 meV, in excellent agreement with the continuum-model results and experimental constraints. Furthermore, We calculated the shift of valence-band relative $\Delta E_v(d)-\Delta E_F(d)$ and the shift of conduction-band relative $\Delta E_c(d)-\Delta E_F(d)$ in Figures S20. The results show that the WS₂ valence band dominates the band modulation. These two points rule out strong hybridization as the main mechanism and instead identify the polar STO(111) surface dipole as the dominant origin of the moiré potential. For full details of the DFT computational framework, registry configurations, and analysis of the band structure and modulation, please see the revised Results section (Figures 4d–f, as shown below) and Methods.

Taken together, the continuum model and DFT calculations provide a quantitative basis for our interpretation. The observed optical resonances are consistent with well-defined moiré exciton minibands. We hope that this substantial addition of theoretical modeling and supporting data addresses the referee's concern and demonstrates that our conclusions are well grounded in both experimental evidence and rigorous calculations. Building on these theoretical results, which show excellent consistency with the experimental observations, we

have incorporated a new Figure 4 into the revised manuscript and Figures S18–S20 into the Supplementary Materials; the new version of Figure 4 used in the main text is shown below as Figure S5.

Figure R5. Moiré-exciton model and DFT calculation, revised Figure 4 in the main text. (a-b) Moiré-exciton continuum-model miniband dispersions (left) and corresponding optical resonances (right) for twist angles $\theta=1^\circ$ and 6° . (c) Moiré-exciton resonances as a function of twist angle θ . (d-e) Structural and electronic properties of WS₂/STO(111) in the high-symmetry AA stacking configuration. (d) Atomic stacking arrangement, where the AA registry corresponds to a $\sqrt{3} \times$ WS₂ monolayer matched to Ti-terminal STO(111). (e) Corresponding unfolded band structure highlighting the electronic characteristics of the heterostructure. (f) DFT calculations of the bandgap modulation $\Delta(d)$, where d represents relative in-plane displacement between WS₂ and STO. The black solid lines represent the supercell of the WS₂/STO heterostructure, whereas the white dashed lines represent the primitive unit cell of WS₂.

References

1. MacDonald, AH. *et al.* Topological Exciton Bands in Moiré Heterojunctions. *Phys. Rev. Lett.* **118**, 147401 (2017).

Reviewer Comment:

2. Low-temperature PL spectra are entirely absent. These are essential to support the ΔR measurements and claims of discrete quantum states. Furthermore, PL (along with polarization-resolved PL and magneto-optics, as well as power- dependent and excitation energy-dependent PL experiments) could reveal signatures of exciton trapping and linewidth narrowing, helping to differentiate between moiré-induced effects and inhomogeneous broadening due to defects or strain.

Response:

We thank the reviewer for this important point. As detailed in the *General Response* (Figures R1-R3), we now include low-temperature PL that corroborates the DR spectra and supports the assignment of moiré excitons. The PL and DR peaks are energy-aligned and exhibit the same twist-angle dependence. With increasing temperature, the low-energy defect emission is fully quenched by ~100 K, whereas the moiré-exciton peaks persist and follow the band-gap redshift. Spatial PL maps reveal flake-wide uniformity for moiré peaks while the defect emission is localized. Power-dependent and polarization-resolved PL further confirm the excitonic origin and valley polarization/coherence. These measurements rule out defect- or strain-induced inhomogeneous broadening and support the moiré-exciton interpretation. The corresponding figures and text have been added to the Main Text and Supplementary Information. While valuable, excitation-energy-dependent PL and magneto-PL measurements are not supported by our current instrumentation and are beyond the scope of this work. We appreciate the reviewer's recommendation and will pursue them in follow-up studies.

Reviewer Comment:

3. The claim of novelty in forming moiré superlattices between oxides and TMDs is perhaps overstated. Prior reports have demonstrated similar systems. For example, *Mark J Hastrup et al 2023 J. Phys.: Condens. Matter 35 194001* discusses moiré formation at oxide-MoS₂ interfaces. The authors should cite this and clearly describe their technical and conceptual advances.

Response:

We thank the reviewer for bringing up the important relevant work by Hastrup *et al.* (*J. Phys.: Condens. Matter* 35, 194001, 2023), which we now cite and discuss in our revised manuscript. Their study indeed demonstrates moiré pattern formation arising from the epitaxial growth of monolayer MoS₂ on SrTiO₃ single crystal with different crystallographic orientations. This work highlights the influence of substrate symmetry on rotational domain formation in TMDs and points out the emergence of moiré features at SrTiO₃-TMD interfaces. However, we respectfully clarify that our study introduces a conceptually and technically distinct approach to engineering moiré heterostructures:

- I. Dimensionality and tunability: The study by Hastrup *et al.* investigates substrate-aligned systems, where monolayer MoS₂ is epitaxially grown on bulk oxide crystals. In such configurations, moiré patterns arise from lattice mismatch or symmetry-driven domain alignment. However, because the twist angle cannot be deliberately adjusted, the resulting moiré periodicity is fixed, which restricts the ability to dynamically control moiré-driven physical phenomena. In contrast, our strategy leverages quasi-2D or freestanding oxide membranes synthesized by pulsed laser deposition, with thickness controlled down to just a few nanometers, which are then integrated with monolayer TMDs grown by CVD using either wet- or dry-transfer techniques. This hybrid approach enables precise rotational alignment and deterministic layer-by-layer stacking, closely mirroring the van der Waals assembly of 2D materials, offering a

degree of tunability and structural control that is unattainable in conventional epitaxial growth on single crystal substrates.

- II. Sample preparation strategy: Our approach combines PLD-grown ultrathin oxides with CVD-grown monolayer TMDs, assembled through a solution-based transfer technique, establishing a new platform for constructing oxide–TMD twist bilayers. This strategy provides:
 - Versatile twist-angle engineering, enabling tailored moiré pattern design.
 - Clean interfaces with minimal interdiffusion or chemical bonding, preserving the intrinsic properties of both layers.
 - Integration of functional ultrathin oxides (ferroelectric, quantum paraelectric, magnetic, etc.) into moiré systems, opening pathways to novel emergent phenomena.
- III. First demonstration of freestanding oxide–TMD moiré superlattices: To the best of our knowledge, this work presents the first experimental realization of a moiré superlattice formed between an ultrathin oxide membrane and a 2D TMD, with both constituents exhibiting two-dimensional–like electronic and mechanical characteristics. This platform establishes a unique bridge between strongly correlated oxides and van der Waals heterostructures, unlocking opportunities for new physics, including moiré-modulated ferroelectricity, hybrid excitonic states, and tunable correlated phenomena.

We thank the referee for drawing our attention to this pioneering study. In the revised manuscript, we have cited this work and acknowledged its significance in advancing the broader understanding of moiré physics. While our study pursues a distinct direction, we fully recognize the importance of this contribution to the field.

Reviewer Comment:

4. The reported PL spectra at room temperature show a dominant feature at ~ 1.97 eV, with no significant neutral exciton peak at ~ 2.00 eV, as typically observed in high-quality WS₂. This suggests different scenarios: (i) significant carrier doping in the WS₂, possibly due to intrinsic quality or (ii) charge-transfer effects from/to STO (thus, it is a trion). Another possibility would be (iii) screening of the Coulomb interactions from the high- κ dielectric environment (however the latter would result in a blueshift of the exciton, please see <https://pubs.acs.org/doi/full/10.1021/acsnano.4c11563>). The authors should clarify the origin of the main emission energy and discuss its impact on the observed optical features. Furthermore, the fitting presented in Figure S5 appears imprecise, with an unusually broad trion linewidth and a potentially incorrect assignment of the main peak to neutral excitons. A comparison with the reflectivity spectra would help clarify this issue. Notably, the room-temperature excitonic emission is significantly broadened-comparable to, or even worse than, that observed in low-quality TMDs on SiO₂/Si substrates.

Response:

We appreciate the reviewer's careful assessment. To address this point, we examined room-temperature PL and DR for WS₂/SiO₂ and WS₂/STO. (i) As shown in Figure R6a, for as-grown WS₂/SiO₂, the main PL peak is aligned with the DR resonance at ~ 1.95 eV. We therefore attribute this to the neutral exciton X_A, a result that suggests the absence of significant carrier

doping in the pristine flakes. We note that the peak energy is indeed relatively low compared to the exfoliated samples. We realize that one reason for the lower peak energy could be the residual strain generated during the CVD growth process, as previously reported [1]. (ii) For WS₂/STO (Figures. R6b-c), the PL peak is red-shifted from the DR resonance at room temperature, indicating trion emission due to interfacial charge transfer. We note that this phenomenon disappears at low temperatures, which may indicate the presence of a temperature-driven charge transfer effect. (iii) Comparing the two types of samples, the exciton energy of WS₂/STO is blue-shifted relative to the exciton energy of WS₂/SiO₂ (Figures. R6d-e), which is consistent with the recent results on the dielectric screening effect of monolayer WSe₂ in a high- κ environment [2].

Taken together, the room-temperature spectra of WS₂/STO reflect the influence of charge transfer and dielectric screening, whereas the moiré-exciton features discussed in the main text are established from low-temperature measurements, where PL and DR are energy-aligned and charge-transfer effects are minimal. Regarding old Figure S5 (room temperature PL), we agree the original fit was not accurate enough. We have now updated the room-temperature PL analysis with improved constraints and line shapes (see Figure R6d-e). Finally, we would like to thank the reviewer for prompting a more comprehensive consideration of these effects.

Figure R6. Room-temperature PL and DR of monolayer WS₂ on SiO₂/Si and STO. (a) PL (black curve) and DR (red curve) of as-grown WS₂/SiO₂/Si at room temperature. The main resonance at ~1.95 eV is assigned to the neutral exciton X_A⁰. (b, c) Room-temperature PL (b) and DR (c) of WS₂/STO. All spectra are vertically offset for clarity. The exciton energy of WS₂/STO is higher than that of WS₂/SiO₂/Si, consistent with high- κ dielectric screening. (d-e) PL spectra of as-grown WS₂/SiO₂ and WS₂/STO, fitted with WS₂ neutral-exciton (X_A⁰) and trion (X_A⁻) components.

References

1. McCreary, K., Hanbicki, A., Singh, S. *et al.* The Effect of Preparation Conditions on Raman and Photoluminescence of Monolayer WS₂. *Sci Rep* **6**, 35154 (2016).
2. Barbone, M. *et al.* Breakdown of the Static Dielectric Screening Approximation of Coulomb Interactions in Atomically Thin Semiconductors. *ACS Nano* **19**, 4 4269–4278 (2025).

Reviewer Comment:

5. Regarding the observed variation in SHG intensity as a function of twist angle, it is unclear why simply the possibility of isotropic biaxial strain -potentially arising from symmetric moiré-induced buckling in the TMD layer- is not considered or discussed beyond complex

charge transfer processes that sensitively depend on interlayer distance. This scenario could possibly contribute to the SHG modulation and should be addressed by the authors.

Response:

We sincerely thank the referees for their thoughtful suggestions and professional insights. It is well established that strain can strongly influence SHG responses; for instance, under uniaxial strain, the polar-SHG patterns typically evolve from circular to elliptical, as illustrated in Figure R7a. Indeed, angle-dependent SHG under varying strain conditions has been extensively reported and quantitatively analyzed in the literature. In contrast, the nearly circular polar-SHG pattern observed in our WS₂ samples (Figure R7b) indicates that strain effects are negligible and that no significant residual strain is present in our WS₂ layer.

Figure R7. Polarization-resolved SHG intensity pattern. (a) Unstrained and 1.0% tensile strain TMD crystal. Adapted from Lukas Mennel *et al.*, *Nature Communications* **9**, 516 (2018), licensed under a Creative Commons Attribution 4.0 International License (<http://creativecommons.org/licenses/by/4.0/>). (b) Linear polarization dependence of the SHG intensity of WS₂/STO TBL with twist angles of 3° and 33°.

References

1. Mennel, L., Furchi, M.M., Wachter, S. *et al.* Optical imaging of strain in two-dimensional crystals. *Nat Commun* **9**, 516 (2018).

Reviewer Comment:

6. While the manuscript presents broad claims regarding the potential of oxide/TMD moiré systems for quantum materials and correlated phenomena, these statements remain overly general. A more concrete and focused discussion outlining specific applications and/or possible research directions grounded in the demonstrated moiré formation in WS₂/STO heterostructures is necessary to support the broader impact of this work.

Response:

We thank the referee for this valuable comment and suggest that a more concrete discussion strengthens the manuscript. The demonstrated moiré formation in WS₂/STO heterostructures, where STO serves as both a quantum paraelectric and high- κ dielectric, provides a solid foundation for engineering emergent quantum states through interfacial coupling of distinct order parameters. This platform points toward several promising research directions, including moiré-modulated ferroelectric fluctuations, hybrid excitonic phenomena, and moiré-tuned dielectric screening and electronic correlations. Beyond WS₂/STO, these concepts can be extended to a broad class of oxide/TMD combinations. Unlike conventional van der Waals moiré systems, oxide layers introduce unique degrees of freedom—ferroelectricity, magnetism, and strong electronic correlations—into the moiré framework. For instance, coupling ferroelectric oxides such as BaTiO₃ or PbTiO₃ with semiconducting TMDs could allow electric-field control of moiré potential landscapes and excitonic localization, while interfacing correlated oxides like rare-earth nickelates with TMDs may enable moiré-mediated bandwidth- or doping-tunable Mott transitions. Similarly, magnetic oxides offer the possibility of proximity-induced exchange interactions that could generate spin-polarized moiré bands and topological magnetic states. Finally, the inherent electrostatic and mechanical tunability of oxide membranes provides powerful levers for dynamically controlling interlayer coupling and strain. Collectively, our WS₂/STO results demonstrate not only feasibility but also distinctive versatility, positioning oxide/TMD moiré systems as a promising and reconfigurable platform for advancing correlated, topological, and multifunctional quantum phenomena. Since several of the concepts outlined above are already central to our ongoing research, we have distilled their key aspects into the revised Discussion to provide a focused and forward-looking perspective, as follows:

“Looking ahead, the demonstrated moiré formation in WS₂/STO and WS₂/LSMO twisted heterostructures provides a robust foundation for engineering emergent quantum states through interfacial coupling of distinct order parameters. The tunable interfacial moiré potential enables systematic exploration of exciton dynamics, charge transfer, and many-body interactions under well-controlled twist conditions. Beyond the systems examined here, these concepts can be generalized to a broad range of oxide/TMD combinations, where functional oxides such as ferroelectrics, correlated nickelates, and magnetic manganites introduce new opportunities for moiré-modulated ferroelectricity, Mott transitions, and spin-polarized moiré bands. Furthermore, the inherent electrostatic and mechanical tunability of oxide membranes offers powerful means to dynamically control interlayer coupling and strain. Collectively, our results not only establish the feasibility of oxide–TMD moiré heterostructures but also highlight their distinctive versatility as a reconfigurable platform for advancing correlated, topological, and multifunctional quantum phenomena.”

Point-to-point responses: Reviewer #3

Reviewer Comment:

The authors investigate the effects of artificially engineered moiré superlattices formed by stacking and twisting a vdW-WS₂ on complex oxide materials (STO(111) and LSMO(111)), including both a freestanding oxide and thin films grown on substrates. The impact of varying twist angles is explored primarily through electron microscopy and optical characterizations. The topic is timely and potentially impactful, as hybrid heterostructures combining 2D materials with complex oxides could lead to intriguing emergent phenomena. However, the presentation of the work lacks clarity, coherence, and focus. For instance, the study begins with WS₂/freestanding STO and then shifts to WS₂/LSMO thin films on STO substrates without clearly articulating the rationale or connection between these systems. As a result, the work contents are disjointed. Future readers are taken from one experimental setup to another without a well-defined conceptual link or progression.

In addition, the scientific depth is limited (e.g., orbital hybridization and charge transfer mechanisms at the interfaces are only superficially addressed). The methodology lacks twist angle control, which is a key factor in current moiré engineering, and the overall approach does not demonstrate significant innovation beyond standard characterization techniques. It appears that the authors simply identified various twisted regions after spreading WS₂ flakes onto oxide layers, rather than systematically engineering specific twist angles. The experimental findings also do not show substantial novelty compared to twisted 2D/2D (e.g., WSe₂/WSe₂) systems, aside from the change in material class. As such, the conceptual advance and impact of the work are not clearly established. Given these concerns, particularly the lack of methodological rigor, insufficient experimental clarification, and limited scientific insight, I do not find the manuscript suitable for publication in *Nat. Commun.* Below are my comments:

Response:

We are grateful to the referee for recognizing our study as both timely and potentially impactful, and we appreciate his/her thoughtful comments. The comments on coherence, methodological rigor, and theoretical support prompted a major revision that, we believe, clarifies the narrative and substantially strengthens the science. Concretely, we (i) added a continuum moiré-exciton model and first-principles DFT, (ii) expanded low-temperature optical spectroscopy (DR and PL) and introduced ultrafast pump-probe to probe interfacial dynamics, (iii) demonstrated deterministic twist-angle control with a PCL-based dry transfer, and (iv) reorganized the Results to make the progression from WS₂/STO to WS₂/LSMO explicit. A brief summary and figure-level pointers are provided below; a point-to-point reply follows in the response.

1) Coherence and scope (WS₂/STO → WS₂/LSMO).

Our goal is a *generic oxide-TMD moiré platform*, not a single materials combination. We therefore begin with WS₂/STO to establish moiré formation and interfacial optical signatures, and then introduce WS₂/LSMO to *leverage correlated magnetism* that is absent in TMD/TMD stacks. SHG shows a pronounced 60° periodicity on (111) oxides

and is enhanced when a conducting magnetic LSMO layer is inserted, directly tying twist angle to moiré superlattices and interfacial dielectric modulation. Complementary pump–probe reveals twist-dependent charge-transfer amplitudes and sub-ps transfer times, as well as few-ps (orbital–lattice) and hundreds-ps (spin–lattice in LSMO) channels—signatures of spin-polarized injection into majority-spin e_g states. These measurements establish oxide-enabled interfacial coupling in the moiré regime and explain why LSMO is studied alongside STO.

2) Theoretical support (continuum model + DFT).

We now provide quantitative theory that underpins the spectroscopic assignments in the revised manuscript. The continuum moiré-exciton Hamiltonian reproduces multiple bright transitions, the overall redshift at small twist angle (θ), and the emergence of mX_2 for $\theta \lesssim 2^\circ$; fitting the mX_1/mX_2 dispersions yields a moiré-potential depth $V_m = 50 \pm 10$ meV. DFT for $WS_2/STO(111)$ across AA/AB/AC registries finds type-II alignment with the WS_2 valence band fully inside the STO gap (non-hybridized) and only minimal mixing at the WS_2 conduction-band K point; mapping over lateral registry gives a ~ 70 meV band-edge modulation. Together, these results rule out strong interlayer hybridization and identify registry-dependent fields from polar STO(111) as the dominant origin of the moiré potential, consistent with experiment-constrained continuum modeling.

3) Twist-angle control and methodology.

To address the request for deterministic control, we added a PCL-based dry-transfer workflow achieving $\sim 1^\circ$ precision, validated by OM/SHG at targeted angles (0° , 2° , 3° , 5° , 54°). We also explain why wet transfer was used for systematic surveys—offering superior throughput and uniformity—while dry transfer is reserved for cases demanding highest angular precision (e.g., transport).

4) Depth of spectroscopy and interfacial physics.

Low-T DR/PL identify two discrete, twist-tunable resonances (mX_1 , mX_2) on the low-energy side of X_A ; temperature, spatial, power, and polarization systematics confirm K-valley neutral excitons (moiré minibands) rather than defect/strain artifacts. SHG power-law checks and polarization analyses exclude strain as the dominant factor, while the $WS_2/LSMO/STO$ pump–probe data quantify twist-modulated charge- and spin-lattice dynamics at the interface.

5) Generality beyond STO and novelty vs 2D/2D.

The Supplementary now includes WS_2 integrated with multiple perovskite oxides ($LaCoO_3$, $PbZrO_3$, $PbTiO_3$), all showing similar twist-angle-dependent SHG intensity variations. This demonstrates that moiré phenomena and interfacial dielectric modulation persist across perovskite platforms and introduces oxide-specific degrees of freedom (ferroelectricity, magnetism, strong correlations) that are not accessible in TMD/TMD systems.

6) Organization and clarity.

To improve readability, we restructured the Results and Discussion parts, enumerating new experiments, modeling, and controls.

We appreciate the referee’s guidance; it led to substantive additions that transform the narrative

from qualitative to quantitatively substantiated. We hope these revisions address the concerns regarding coherence, rigor, and impact. A detailed, point-by-point reply follows below:

Reviewer Comment:

1. What is the role of complex oxide materials here? While the abstract and introduction suggest that electronic interactions at the WS₂/oxide interfaces are central to the work, the manuscript lacks a detailed investigation into these interfacial effects. In particular, there is no in-depth analysis of atomic-level interactions around the interface or electronic reconstruction (e.g., orbital hybridization, charge transfer), which are expected when coupling weakly correlated vdW materials with strongly correlated complex oxides. To support the claims and deepen the physical understanding, the authors could perform DFT calculations to examine band structure evolution and flat band formation as a function of twist angle (see this example, *Nat. Commun.* 12, 5601 (2021)) for relevant methodology. This would also clarify whether the oxide layers contribute to moiré-induced flat bands. Given that one of the key motivations stated in the abstract and introduction is to explore the interplay between vdW and correlated oxide systems, I strongly recommend that the authors include a more rigorous theoretical or spectroscopic investigation to resolve interfacial orbital interactions and emergent phenomena, which are not well understood in the presented work.

Response:

We sincerely thank the referee for raising these critical questions, which touches on the central motivation of our work. The comments prompted us to carefully re-examine the role of complex oxides at the WS₂/oxide interface and to undertake substantial revisions of the manuscript. In response, we have carried out new theoretical calculations and expanded spectroscopic investigations, all of which directly address the referee's concerns. We are grateful for this opportunity, as the question has significantly strengthened and refined the clarity, depth, and completeness of our study. Since the referee's comments span several important aspects, we therefore provide a point-by-point response below:

1) Role of complex oxide.

Our central finding is that perovskite oxides are indeed impactful functional layers; their correlated and polar surface chemistry imprints a deep, twist-tunable moiré potential onto the TMD. On STO(111), the polar stacking produces a registry-dependent interfacial field that strongly modulates the WS₂ band edges and, in turn, the exciton landscape and charge-transfer pathways. This oxide–TMD coupling manifests as twist-angle–dependent exciton minibands and interfacial charge/spin dynamics, establishing complex oxides as active knobs for moiré engineering rather than inert substrates. The new continuum-model and DFT results presented in the revised manuscript (primarily in Figure 4) provide critical insights into the oxide–TMD interactions. *These findings, together with the optical and dynamical measurements, are substantiated and elaborated throughout the revised Results section, with comprehensive supporting evidence in Figures 3–5.*

2) Direct evidence of interfacial electronic reconstruction (hybridization, charge transfer) via DFT and spectroscopy.

In the revised manuscript, our optical spectroscopy reveals two discrete, twist-tunable resonances (mX_1 , mX_2) on the low-energy side of X_A in low-temperature DR/PL spectra. Additional temperature-, spatial-, power-, and polarization-dependent measurements confirm that these features correspond to K-valley neutral excitons, i.e., moiré minibands, whereas the lower-energy X_D is identified as defect-bound. Complementing these observations, new first-principles DFT calculations of $WS_2/STO(111)$ (Ti-terminated) across local stackings (AA/AB/AC) show a type-II band alignment: the WS_2 valence band lies entirely within the STO gap and remains non-hybridized, while the K point of WS_2 conduction band exhibits only minimal mixing with STO states despite energetic overlap. Mapping the bandgap modulation over lateral registry yields a peak-to-peak amplitude of ~ 70 meV, and Fourier analysis of $\Delta E_g(r)$ produces first-shell components ($V = 5.07$ meV, $\psi = 58.23^\circ$) in close agreement with continuum-model results constrained by experiment (5.55 meV and 60°). Taken together, these optical and theoretical results demonstrate that the dominant moiré potential at the oxide–TMD interface originates from registry-dependent interfacial fields of the polar STO(111), rather than from strong interlayer hybridization. *For detailed changes and explanations, please refer to the revised Results section and Supplementary Notes, where these new optical and theoretical findings are presented alongside updated Figures 3–4.*

3) Continuum modeling and its agreement with experiment.

In the revised manuscript, we constructed a continuum moiré-exciton Hamiltonian (including kinetic and exchange terms with a lowest-harmonic moiré potential) and computed minibands/absorption versus angle. The model reproduces (i) multiple bright transitions and an overall redshift at small θ , and (ii) the emergence of mX_2 for $\theta \lesssim 2^\circ$, matching the measured spectra. By fitting mX_1/mX_2 dispersions, we extract a moiré-potential depth $V_m = 50 \pm 10$ meV, which is consistent with and complementary to the DFT-derived band-edge modulation map. *For detailed derivations, parameter choices, and the corresponding figure-level evidence, please see the revised Results section (Figures 4a–c, 3g) and Methods.*

4) Do oxides contribute to moiré-induced (electronic) flat bands?

Our continuum treatment targets exciton minibands and quantitatively captures the observed optical resonances; DFT shows substantial single-particle band-edge modulation (~ 70 meV) driven by registry-dependent interfacial fields on STO(111), indicating a strong periodic potential at the electronic level. While direct observation of electronic flat bands requires momentum-resolved probes, our results delineate the mechanism and provide clear targets for ARPES/STM/KPFM in future work.

5) Spectroscopies revealing interfacial coupling in TMD/magnetic oxide twisted heterostructures

In the revised manuscript (Figure 5), SHG mapping, highly sensitive to the dielectric environment, reveals a pronounced 60° periodicity in ΔSHG versus twist angle on (111) oxides, with the effect further amplified by the presence of an intervening magnetic LSMO layer. Polarization-dependent SHG retains the intrinsic D_{3h} symmetry of WS_2 , thereby excluding strain as the primary driver and directly linking twist angle to moiré superlattice formation and interfacial dielectric modulation. Complementary ultrafast pump–probe measurements show

that $\Delta T/T$ amplitudes increase and sub-picosecond transfer times shorten with decreasing θ , indicating more efficient interfacial charge transfer at larger moiré length scales. At longer delays, two decay components emerge: a few-ps orbital–lattice recovery and a hundreds-ps spin–lattice relaxation in LSMO, both exhibiting clear twist dependence. These dynamics are consistent with spin-polarized electron injection into majority-spin e_g states, underscoring how twist angle governs charge-transfer efficiency and spin–lattice coupling in TMD/magnetic oxide heterostructures.

6) A generic platform for twisted TMD/oxide heterostructures.

In line with the referees' suggestions, the revised Supplementary Materials now include SHG mappings of WS_2 integrated with multiple perovskite oxides ($LaCoO_3$, $PbZrO_3$, $PbTiO_3$). In all these systems, we observe similar twist-angle–dependent moiré patterns and SHG intensity variations (Figure S21), underscoring the generality of the oxide–TMD moiré platform and its potential to harness strongly correlated phenomena absent in conventional TMD-only stacks.

To briefly summarize for the referee, we have added a rigorous theoretical–experimental package that directly addresses the referee's requests. Continuum modeling reproduces the twist-angle evolution of mX_1/mX_2 and yields $V_m = 50 \pm 10$ meV, while DFT maps a ~ 70 meV registry-dependent band-edge modulation and shows minimal hybridization (type-II alignment), identifying the polar STO(111) interfacial field as the primary origin of the moiré potential. Complementary temperature-, spatial-, power-, and polarization-dependent PL measurements, SHG, and ultrafast pump–probe measurements establish twist-controlled charge transfer and spin–lattice dynamics—clear signatures of oxide-enabled interfacial coupling in the moiré regime. We hope that these comprehensive responses and additional results satisfactorily address the referee's concerns.

Reviewer Comment:

2. In lines 124–126, the authors state that the moiré pattern becomes smeared as the thickness of the bottom layers increases. However, they do not provide a physical explanation, references, or experimental support for this statement. Additional discussion is needed to clarify the underlying mechanism (whether it relates to diminished interfacial coupling, increased surface roughness, strain relaxation, or other effects in thicker films). Then, the authors should explicitly explain how this observation motivated their use of ultrathin oxide layers in the study.

Response:

Thank you for pointing this out. We regret any confusion caused by our earlier wording. To clarify, our statement does not imply that the moiré superlattice/pattern actually disappears when the bottom oxide layer becomes thicker. Rather, the visibility of moiré patterns in TEM strongly depends on the thickness of the constituent layers. As the thickness of the bottom layer increases, the moiré contrast becomes progressively smeared due to several mechanisms[1]. As the electron beam passes through a thicker specimen, the signal becomes affected by stronger multiple scattering, increased background contrast, and averaging over many atomic planes. These effects blur the interference fringes that define the moiré pattern, making the contrast appear smeared even though the periodicity is still present. Projection blurring and

contrast delocalization in thicker films obscure fine periodic modulations and the reciprocal-space visibility of moiré satellites is also suppressed in thicker samples, since multiple scattering reduces the intensity of superlattice reflections.

Thus, the moiré superlattice is still physically present, but its TEM visibility is diminished by thickness-related imaging constraints. This limitation directly motivated our use of ultrathin freestanding oxide layers, which minimize thickness-induced degradation and allow clear, high-fidelity visualization of moiré interference for reliable analysis of interfacial physics. We have removed previous lines to avoid possible misunderstanding.

References

1. Williams, D. B. & Carter, C. B., *Transmission Electron Microscopy: A Textbook for Materials Science Springer* **2nd** Ed., (2009).

Reviewer Comment:

3. It is unclear how precisely the twist angles presented in Fig. 3 (actually Fig. 4) were controlled or determined. For clarity, the authors should include representative images or spectral maps corresponding to the different twist angles that give rise to varied optical responses.

Response:

We appreciate the reviewer's request for clarity. As explained in the main text, we define the 0° twist configuration when the zig-zag edge of triangular WS_2 aligns parallel to the $[11\bar{2}]$ direction of $\text{STO}(111)$, which yields the largest moiré periodicity. Based on this definition, the twist angle of each WS_2 flake was determined from the azimuth of the flake edge relative to the $\text{STO} [11\bar{2}]$ direction within the same imaging frame (to avoid inter-sample errors). Our initial estimate was based on a single equilateral edge. Following the reviewer's suggestion, we now measure all three edges separately and report θ as the average of the three edge azimuths, with uncertainty given by the standard deviation. The resulting angles are $0.2 \pm 0.2^\circ$, $1.0 \pm 0.1^\circ$, $1.7 \pm 0.1^\circ$, $6.2 \pm 0.1^\circ$, and $7.9 \pm 0.8^\circ$, as shown in Figure R8. We have updated all angle labels in the manuscript accordingly.

Figure R8. Optical microscopy (OM) images and twist-angle definition. (a) Schematic defining the twist angle θ as the azimuth of the WS₂ flake edges relative to the STO [112] direction. (b-f) OM images of representative WS₂ flakes with $\theta=0.2^\circ$, 1.0° , 1.7° , 6.2° , and 7.9° , respectively.

Reviewer Comment:

4. This manuscript would benefit from including atomic force microscopy (AFM) and X-ray diffraction (XRD) 2θ scans of both the freestanding single-crystalline oxide layers on Si and the grown thin films on STO in SI as they can support for the claimed abrupt interface engineering.

Response:

We thank the reviewer for this thoughtful suggestion. As suggested, atomic force microscopy (AFM), X-ray diffraction (XRD) 2θ scans, and ϕ (phi) scans for both the freestanding single-crystalline oxide layers (Figure R9a & Figure R9b) on Si and the epitaxially grown thin films on STO (Figure R9c & Figure R9d) have been incorporated in the revised Supplementary Information. These characterizations confirm the high crystalline quality, smooth surfaces, and the epitaxial relationship across the interfaces, thereby supporting the claim of abrupt interface engineering. For clarity, these data are also presented directly below this section.

Figure R9 shows the (a) 2 theta scan STO/LSMO/STO(111) and (b) Phi scan of STO/LSMO/STO111. Fig (c) & (d) shows the quality of as grown STO(111) before and after freestanding remains the same of high quality.

Reviewer Comment:

5. Regarding the emergence of two discrete excitonic quantum states (Fig.3), I am curious whether the authors have also tested with the case of $WS_2/LSMO$. Also, have the authors investigated the temperature dependence of these excitonic states? Temperature-dependent measurements could help confirm their quantum-confined nature and distinguish them from other possible spectral features such as defect-bound excitons or inhomogeneous broadening effects.

Response:

We thank the reviewer for the helpful suggestions. As detailed in the General Response and incorporated into the revision, temperature-dependent PL shows that the moire exciton peaks persist and follow the band-gap redshift with increasing temperature, whereas the low-energy defect emission is fully quenched by ~ 100 K. Regarding $WS_2/LSMO$, measurements are currently in progress but remain inconclusive, and the related findings will be presented in full in future work.

Reviewer Comment:

6. What is the thickness of STO and LSMO layers used in this work? Is there any thickness dependence? I see the PLD growth conditions used in this work for both films are rather

unconventional (e.g., 10 Hz). I am curious whether all the grown films were epitaxial.

Response:

We thank the reviewer for this insightful question. In this work, the STO (LSMO) layers were grown with nominal thicknesses of approximately 4 nm (~10 unit cells) and 2 nm (~5 unit cells), respectively. The ultrathin LSMO layer serves a dual role: it maintains epitaxial registry with the underlying oxide substrate and acts as a charge reservoir, facilitating interfacial charge transfer to WS_2 and thereby enabling modulation of the local electronic environment in the moiré superlattice. While we have not yet performed a systematic study of thickness dependence, the chosen dimensions fall within the ultrathin regime, where interfacial coupling and proximity effects are expected to be particularly evident.

The reviewer is correct regarding the influence of repetition rate on the growth rate in PLD. Indeed, for ultrathin film growth, slower repetition rates (such as 1–3 Hz) can help to effectively control the deposition rate and thereby provide fine tuning of film growth[1-3]. However, careful optimization of PLD growth parameters shows that both slower repetition rates and faster ones, such as 10 Hz, can produce films with equally high epitaxial quality. In our optimized conditions, even at 10 Hz the films consistently exhibit smooth morphology and maintain high-quality single-crystalline epitaxy. The epitaxial quality of both STO and LSMO layers was confirmed by X-ray diffraction, which demonstrated coherent in-plane lattice/symmetry matching and high crystalline quality (see Figure R10). Furthermore, we also performed additional XRD characterizations on other pseudo-cubic perovskite oxides, such as $LaCoO_3$ and $SrMnO_3$, grown on (111)-oriented STO substrates under similar conditions, and found that all films exhibited clear epitaxial relationships (see Figure R11). These growth conditions are also consistent with prior studies on ultrathin manganite and titanate films, underscoring the reproducibility and reliability of our approach [4-8].

Figure R10 shows the 2 theta scan (a)LSMO/STO(111), (c) STO/LSMO/STO(111) and Phi scan of (b) LSMO/STO (111), (d) STO/LSMO/STO111.

Figure R11 shows the 2 theta scan (a)LCO/STO(111), (c) SMO/STO(111) and Phi scan of (b) LCO/STO (111), (d) SMO/STO111.

References

1. Shewale, PS., Yu, YS. The effects of pulse repetition rate on the structural, surface morphological and UV photodetection properties of pulsed laser deposited Mg-doped ZnO nanorods. *Ceramics International* **42**, 7125-7134 (2016).
2. Zhaoyang, Wang *et al.* Effect of laser repetition frequency on the structural and optical properties of ZnO thin films by PLD. *Vacuum* **85**, 397-399 (2010).
3. Gabriel, V., Kocán, P., Bauer, S. *et al.* Effect of pulse laser frequency on PLD growth of LuFeO₃ explained by kinetic simulations of in-situ diffracted intensities. *Sci Rep* **12**, 5647 (2022).
4. Dong, Z., Zhang, Y., Chiu, CC. *et al.* Sub-nanometer depth resolution and single dopant visualization achieved by tilt-coupled multislice electron ptychography. *Nat Commun* **16**, 1219 (2025).
5. Lin, CY., Chen, BC., Liu, YC. *et al.* Integration of freestanding hafnium zirconium oxide membranes into two-dimensional transistors as a high- κ ferroelectric dielectric. *Nat Electron* **8**, 560–570 (2025).
6. Wu, PC., Wei, CC., Zhong, Q. *et al.* Twisted oxide lateral homostructures with conjunction tunability. *Nat Commun* **13**, 2565 (2022).
7. Staub, Urs. *et al.* Antiferrodistortive and Ferroelectric Phase Transitions in Freestanding Films of SrTiO₃. *Nano Lett.* **25**, 7651–7657 (2025).
8. Chuang, YD. *et al.* Presence of Delocalized Ti 3d Electrons in Ultrathin Single-Crystal SrTiO₃. *Nano Lett.* **22**, 1580–1586 (2022).

Reviewer Comment:

7. In Fig. 5, the authors describe a comparison with “various complex oxides,” but only two oxides (STO and LSMO) are included, while SiO₂ is not a complex oxide. Therefore, the term “various” seems overstated in this context. Furthermore, all presented systems exhibit SHG signals, but it is unclear whether the authors performed control experiments on bare STO and SiO₂ substrates without WS₂. Both substrates are known to potentially exhibit non-trivial SHG responses due to surface or defects or local symmetry effects, so attributing SHG solely to moiré superlattices requires stronger clarification. To support their interpretation, the authors need to include angle-dependent SHG measurements of the bare substrates (STO and SiO₂). This would help confirm whether the SHG signals observed in WS₂/oxide stacks indeed originate from moiré-induced effects rather than intrinsic substrate contributions.

Response:

We thank the referee for this constructive input and agree that including additional oxide systems provides a more convincing demonstration of the generality of our approach. To further demonstrate that the observed phenomena are not limited to STO and LSMO but represent general features across complex oxides, we provide in Figure R12 both schematic and SHG images of WS₂ transferred onto several additional (111)-oriented complex oxides, including LaCoO₃, PbZrO₃, and PbTiO₃. It is remarkable that similar moiré patterns and SHG intensity modulations are observed in WS₂/LaCoO₃, WS₂/PbZrO₃, and WS₂/PbTiO₃ heterostructures, confirming the reproducibility of the moiré superlattice effect across different oxide platforms. These examples suggest that when the selected oxides have pseudo-cubic perovskite structures with comparable lattice symmetry and mismatch, the resulting WS₂/oxide heterostructures consistently form moiré superlattices.

Additionally, we explicitly include the case of WS₂ on ferroelectric PbTiO₃ (111), which is known for its strong intrinsic SHG response. Unlike WS₂/LCO(111) and WS₂/PZO(111), the SHG signal from WS₂/PTO(111) displays classical interference behavior, where constructive and destructive interference between the SHG signals of WS₂ and PTO governs the overall response [1,2]. As shown in Figure R12k, destructive interference nearly extinguishes the SHG signal, while in Figure R12l, constructive interference enhances the SHG intensity well beyond that of WS₂ alone. These results highlight the distinct mechanisms at play in WS₂/PTO compared to other WS₂/oxide heterostructures, while reinforcing that moiré-induced SHG phenomena are broadly reproducible across multiple complex oxides.

Figure R12. Schematic illustrations and SHG images of WS₂/LaCoO₃ (a-d), WS₂/PbZrO₃ (e-h), and WS₂/PbTiO₃ (i-l). Scale bars for moiré illustration and SHG are 5 nm and 20 μ m, respectively.

References

1. Urbaszek, B. *et al.* Second harmonic generation control in twisted bilayers of transition metal dichalcogenides. *Physical Review B* **105**, 115420 (2022).
2. Chang, WH. *et al.* Second harmonic generation from artificially stacked transition metal dichalcogenide twisted bilayers. *ACS nano* **8**, 2951-2958 (2014).

Following the referee's suggestion, we conducted additional SHG measurements on monolayer WS₂ as well as on the corresponding bare substrates, including SiO₂, STO, and LSMO, as shown in Figure R13. In contrast to the strong response from WS₂, the SHG signals from these oxide substrates were extremely weak by comparison, indicating that their contribution to the overall response is negligible. This confirms that the SHG intensity modulations observed in WS₂/oxide heterostructures arise predominantly from moiré-induced effects, rather than from the intrinsic nonlinear optical properties of the substrates. *Figure R13 has been incorporated into the revised Supplementary Materials.*

Figure R13 SHG spectra of WS₂/Si, WS₂/LSMO, WS₂/STO, along with the corresponding bare substrates (Si, LSMO, and STO) for comparison.

8. Are there any established strategies to precisely control twist angles between two-dimensional (2D) materials or FS membranes? If so, have the authors applied one of the methods in this work?

Response:

We thank the referee for this insightful question. Indeed, several established methods exist for controlling the twist angle in two-dimensional (2D) heterostructures, each with distinct advantages and limitations [1–5]. Among these, the “tear-and-stack” approach is the most widely employed for precise angle alignment [1]. In our study, however, we primarily adopted a wet-chemical etching technique [6] to transfer 2D materials onto oxide films and membranes. This method offers clear advantages in terms of time efficiency, cost-effectiveness, and scalability, enabling the fabrication of heterostructures with diverse twist angles and excellent uniformity across the same substrate. Unlike mechanical stacking, it does not require complex manual alignment or specialized equipment, making it highly suitable for systematic investigations of moiré physics. Although its angular precision is somewhat lower than that of tear-and-stack, this approach provides a practical balance between throughput and accuracy,

achieving an estimated twist-angle precision of $\sim 1^\circ$, which is sufficient for exploring moiré phenomena.

Motivated by the referee's question, we carried out a dry-transfer approach of WS_2 flakes onto (111)-oriented oxide thin films using a deterministic PCL-based polymer stamp to demonstrate precise twist-angle control between 2D materials and oxides. The corresponding dry-transfer results have also been included in the Supplementary Materials. The transfer stamp was prepared by coating a PDMS block with a thin PCL layer. A WS_2 flake exfoliated on Al_2O_3 was aligned to the desired twist angle, picked up with the heated stamp, and subsequently released onto a targeted STO region under controlled heating and cooling to minimize wrinkles or fractures. Residual PCL was removed by immersing the sample in tetrahydrofuran overnight. A schematic illustration of the procedure is provided in Figure R14.

Fig. R14. Schematic illustration of the PCL-based dry-transfer process. (a) Alignment of the WS_2 flake orientation to 0° . (b) Rotation of the WS_2 flake to the desired twist angle, followed by heating prior to engaging the PCL stamp. (c) Contact of the PCL stamp with the flake, cooling to room temperature, and subsequent pick-up. (d–e) Alignment of the STO substrate to 0° and heating for the drop-down transfer.

As revealed in Figures 1 and 2 of our manuscript, the twist angle can be efficiently determined from the zig-zag crystallographic direction of WS_2 relative to STO $[11\bar{2}]$, which is parallel to one of the substrate edges of (111)-oriented STO . In Figure R15 optical microscope images show WS_2 flakes transferred onto $\text{STO}(111)$ at twist angles of 0° , 2° , 3° , 5° , and 54° , with corresponding SHG maps confirming the distinct angle-dependent responses. Figure R16 also presents the linear polarization dependence of SHG intensity for the 0° , 2° , and 3° cases, further demonstrating the isotropic symmetry of WS_2 . Together, these results highlight that our deterministic PCL-based dry-transfer method achieves angular precision of $\sim 1^\circ$ and enables proof-of-concept fabrication of twisted WS_2 /oxide heterostructures. If precise twist-angle control of WS_2 on freestanding membranes is desired, an additional freestanding process can

be applied following the PCL-based dry-transfer method [7,8]. In this work, however, we adopted the wet-etching transfer technique for WS₂ on freestanding oxide membranes, rather than the dry-transfer method with precise angle control, as it offers a more efficient route for systematic investigations across a broad range of twist angles.

Overall, while the dry-transfer approach offers higher angular precision, it is inherently more time-consuming and less scalable. In contrast, the wet-etching method is better suited for systematic and reproducible investigations of moiré effects. Together, these complementary approaches underscore the trade-off between precision, scalability, and cost: wet transfer provides a practical and efficient route for exploring a broad range of twist angles under consistent conditions, whereas dry transfer is particularly valuable for transport-based nanoelectronic devices that demand precise twist-angle control.

Figure R15 shows the dry transfer of WS₂ flakes onto oxide substrates using a deterministic dry-transfer setup with rotational control under optical microscopy and their respective SHG images. This approach allowed us to manually align the flakes with similar angular precision ($\sim 1^\circ$).

Figure R16 shows the (a-c) SHG mapping with their respective (d-f) polar graphs with angles.

References

1. Cao, Y., Fatemi, V., Fang, S. *et al.* Unconventional superconductivity in magic-angle graphene superlattices. *Nature* **556**, 43–50 (2018).
2. Castellanos-Gomez, A. *et al.* Deterministic transfer of two-dimensional materials by all-dry viscoelastic stamping. *2D Mater.* **1**, 011002 (2014).
3. Quellmalz, A., Wang, X., Sawallich, S. *et al.* Large-area integration of two-dimensional materials and their heterostructures by wafer bonding. *Nat Commun* **12**, 917 (2021).
4. Yankowitz, M., Xue, J., Cormode, D. *et al.* Emergence of superlattice Dirac points in graphene on hexagonal boron nitride. *Nature Phys* **8**, 382–386 (2012).
5. Li, H. *et al.* Epitaxial growth of two-dimensional layered transition-metal dichalcogenides: Growth mechanism, controllability, and applications. *Chem. Soc. Rev.* **47**, 4581–4593 (2018).
6. Suk, J. W. *et al.* Transfer of CVD-grown monolayer graphene onto arbitrary substrates. *ACS Nano* **5**, 6916–6924 (2011).
7. Kim, Kyoungwan. *et al.* van der Waals heterostructures with high accuracy rotational alignment. *Nano letters* **16**, 1989–1995 (2016).
8. Son, Suhan. *et al.* Strongly adhesive dry transfer technique for van der Waals heterostructure. *2D Materials* **7**, 041005 (2020).

Summary of the revisions:

1. Quantified the possible biaxial-strain contribution using established SHG-strain calibrations and Raman measurements, showing the potential biaxial strain in our work is not sufficient to account for the observed ~25% periodic SHG modulation. These results have been added to the Supplementary Information as suggested by Reviewers 1 and 2.
2. As suggested by Reviewer 3, the abstract, introduction, and discussion sections have been revised to improve clarity and to better highlight the significance and emerging phenomena associated with these oxide–TMD heterostructures.
3. Updated the reference frame of Fig. S5c with respect to STO(111) to more accurately reflect the structural correspondence between the freestanding and as-grown STO films.
4. Added simulations of moiré formation for various TMD/oxide combinations (Fig. R4 and Fig. S29) and performed SHG measurements on MoS₂ transferred onto STO(111) (Fig. R5), demonstrating that moiré-induced modulation also occurs in TMDs beyond WS₂.
5. The thickness of the LSMO layer has been included in the manuscript.

Reviewer 1&2

Comments:

The authors have provided a comprehensive and well-structured rebuttal that clearly reflects a major revision effort. I appreciate the significant amount of new experimental and theoretical work that has been now included into the revised manuscript. The inclusion of low-temperature PL studies, power and temperature dependencies, spatial uniformity analysis, and polarization-resolved measurements provides a much more complete and convincing picture of the excitonic features in these oxide-TMD heterostructures. The addition of continuum modeling and DFT calculations gives quantitative support to the moiré potential estimation and strengthens the overall interpretation. These new results, together with the expanded discussion on the role of complex oxides and improved methodological clarity (particularly regarding twist-angle control), considerably enhance both the scientific depth and coherence of the work.

Overall, I find that the authors have satisfactorily addressed most of the critical points raised in the initial review. The revised manuscript now presents a compelling and well-substantiated case for the formation of moiré excitons at oxide-TMD interfaces and demonstrates the broader potential of this hybrid materials platform.

Response:

We sincerely thank the reviewers for the very encouraging and insightful assessment. We are truly grateful to both referees for their constructive and thoughtful comments in the initial review, which have greatly helped us to improve the scientific depth and overall quality of this work. Their valuable feedback has been essential in refining our analyses, strengthening the interpretation, and bringing the manuscript to its present form.

Comments:

That said, I am not fully satisfied with the response to my comment concerning the SHG intensity modulation (previous comment 5). In their reply, the authors discuss uniaxial strain and use the six-lobe symmetry of the SHG pattern to exclude strain effects. However, my original concern referred to isotropic biaxial strain, which can modulate SHG intensity without altering the symmetry of the polar pattern. This effect has been quantified, for example, in Xing, H., Liu, J., Zhao, Z. et al. "Quantifying the in-plane strain influence on second harmonic generation of molybdenum disulfide" *Commun Phys* 7, 382 (2024), (see Fig. 3c). Given this, it remains unclear why the authors attribute the SHG intensity modulation solely to charge-transfer effects and do not consider the possibility that isotropic biaxial strain -possibly arising from isotropic buckling within the moiré supercell could play a role. Aside from this remaining ambiguity regarding the SHG interpretation, I find the revision convincing and the manuscript substantially improved. The new data provide deeper insight and significantly strengthen the claims, and the study now reaches the level of rigor and completeness expected for publication in *Nature Communications* once this last issue is properly clarified.

Response:

We appreciate the reviewers' insightful comment regarding the possible contribution of isotropic biaxial strain to the observed SHG modulation, which has motivated us to further analyze and elaborate on this aspect in the revised manuscript. As the reviewer pointed out, a biaxial strain can modulate SHG intensity without altering the six-fold symmetry of the polar pattern. In our measurements, the SHG intensity modulation amplitude is approximately 25%. According to the reference suggested by the reviewer [1] (Xing et al., *Commun. Phys.* 7, 382 (2024)), such modulation would correspond to roughly 0.08% biaxial strain.

To evaluate this possibility, we further performed Raman spectroscopy on the same samples. According to the report by Gao's group [2], as shown in Fig. R1(a) and (b), the biaxial strain of 0.51% would cause a shift of $\sim 5 \text{ cm}^{-1}$ in the E^1_{2g} phonon mode. However, as shown in Fig. R1(c), our Raman spectra exhibit no discernible shift in the E^1_{2g} peak for twist angles ranging from 3° to 60° . This result indicates that the strain in our samples is estimated to be $<0.03\%$, as inferred from the trends in Fig. R1(a) and (b). Such a small strain would generate an SHG modulation of $<9.3\%$, which is substantially lower than the $\sim 25\%$ periodic modulation observed in our measurements. Therefore, although we cannot fully rule out the presence of minor biaxial strain, its magnitude is insufficient to explain the pronounced periodic SHG modulation observed in our measurements.

Figure R1 (a) Biaxial strain-dependent Raman spectra of monolayer WS_2 under 532 nm excitation. (b) Biaxial strain evolution of the A_{1g} and $2\text{LA(M)} + E^1_{2g}$ Raman modes, showing an approximate 2 cm^{-1} shift at $\sim 0.08\%$ biaxial strain. Panels (a) and (b) are adapted from Shrvan Roy *et al.*, *Scientific Reports* **14**, 3860 (2024), licensed under a Creative Commons Attribution 4.0 International License (<http://creativecommons.org/licenses/by/4.0/>). (c) Raman spectra of twisted $\text{WS}_2/\text{LSMO}(111)$ heterostructures at twist angles of 3° , 33° , and 60° .

The symmetric SHG pattern and the unshifted Raman modes indicate that the overall averaged strain in the sample is nearly zero. In TMD moiré systems, spatially varying intralayer strain fields can indeed arise, especially at small twist angles and/or when structural reconstruction occurs, which relates to the moiré-induced buckling issue highlighted by the reviewer [3]. However, these strain fields typically alternate between compressive and tensile components within each moiré period. When probed with a micron-sized Raman/SHG spot, such rapidly oscillating components average out, rendering the local strain variations inaccessible to the

measurement. As a result, conventional Raman and SHG techniques generally cannot directly reveal these moire-induced strain modulations. For this reason, we tentatively exclude strain-related effects as the main cause of the observed SHG intensity variation. We sincerely appreciate the reviewer for raising this important point, which allows us to provide a clearer and more balanced discussion. These considerations and **Figure R1c** have been incorporated into the revised manuscript.

References

- [1] Xing, H., Liu, J., Zhao, Z., He, X., & Qiu, W. Quantifying the in-plane strain influence on second harmonic generation of molybdenum disulfide. *Communications Physics* **7**, 382 (2024).
- [2] Roy, S., Yang, X., & Gao, J. Biaxial strain tuned upconversion photoluminescence of monolayer WS₂. *Scientific Reports* **14**, 3860 (2024).
- [3] Wang, J., Tosatti, E. Universal moiré buckling of freestanding 2D bilayers. *Proc. Natl. Acad. Sci. U.S.A.* **121**, e241839012 (2024).

Reviewer 3

Comments:

I noticed that the authors have made great efforts and have satisfactorily addressed all of my previous comments. The revised manuscript has been significantly improved, particularly in the organization and logical connection of the content, as well as in the strengthened theoretical discussion on the moiré potentials and electronic band-gap modulations as a function of twisting angle. These revisions provide a comprehensive understanding of the twist-angle-dependent atomic modulations at the surfaces of complex oxides and the resulting emergent physical phenomena, such as moiré excitons and charge-transfer effects. I also think this work will be of broad interest to both the oxide and 2D-materials research communities. Therefore, I am pleased to recommend publication after addressing the following comments below.

Response:

We sincerely thank the reviewer for the very positive and encouraging evaluation of our revised manuscript. We are grateful that the reviewer finds our responses satisfactory and appreciates the major improvements in the organization, theoretical framework, and clarity of the moiré-related physics. We also deeply appreciate the reviewer's indication of a positive recommendation, which is highly motivating for us. The reviewer's insightful comments throughout the review process have greatly strengthened this work and helped us refine both the experimental interpretations and theoretical discussion. We are pleased that the revised manuscript can now better serve both the oxide and 2D materials research communities, and we truly appreciate the reviewers' constructive guidance in shaping this outcome.

Comments:

1. In the abstract (and also at the end of the Introduction), the authors should include another emergent phenomenon (the twist-angle-dependent symmetry breaking and highly tunable ultrafast charge-transfer effect arising in the magnetic oxide-WS₂ system).

Response: We are grateful for the reviewer's thoughtful suggestion. We have revised the manuscript accordingly and highlighted all relevant revisions suggested by the referee.

Comments:

2. Please indicate the thickness of the LSMO films used in this work in both the main text and the Experimental section.

Response:

We appreciate the reviewer's valuable suggestion to enhance the clarity of the manuscript. The thickness of LSMO used has been explicitly indicated in both the main text and the experimental section. Specifically, the LSMO layer thickness is 2 nm. This sample information has been included accordingly.

Comments:

3. Please clarify why the thickness of the freestanding STO membrane was chosen to be 4 nm rather than a thicker one.

Response:

We appreciate the reviewer's question regarding the choice of a 4-nm freestanding STO membrane. We would like to clarify that this ultrathin STO layer is used exclusively for TEM characterization, where the WS_2 /STO moiré pattern is directly imaged. All optical, SHG, and excitonic measurements of WS_2 /STO in the manuscript are performed on WS_2 interfaced with bulk STO single-crystal substrates with well-defined in-plane $[1\bar{1}2]$ and $[1\bar{1}0]$ crystallographic directions, not on the freestanding membranes.

For TEM observations, maintaining both layers within a comparable thickness range is essential for preserving moiré visibility. When one material is significantly thicker than the other, the moiré contrast becomes progressively smeared: the diffraction signal is dominated by the thicker crystal, while the much weaker contribution from the thinner layer becomes difficult to resolve [1]. This limitation, arising from multiple scattering, projection averaging, and contrast delocalization in thicker specimens, is well known in TEM moiré imaging. Therefore, to avoid these effects, we employ an ultrathin freestanding STO membrane so that the diffraction contributions from WS_2 and STO remain comparable. This ensures that the moiré periodicity and atomic-scale structural modulations can be clearly resolved in the TEM images.

References

1. Williams, D. B. & Carter, C. B., *Transmission Electron Microscopy: A Textbook for Materials Science Springer* 2nd Ed., (2009).

Comments:

4. Please verify whether Fig. S5c is correctly presented. The peak positions and assignments appear inconsistent with those in Figs. S5a,b.

Response:

We thank the reviewer for this insightful observation. In the original supplementary information figure, the L-scan of the freestanding STO (FS-STO) on Si (Fig. S5c) was referenced to the lattice constant of the Si substrate when converting the out-of-plane reciprocal vector (q_z) into L-values. In contrast, the STO substrate and the as-grown ultrathin STO film (Fig. S5a, b) were referenced to the STO(111) lattice spacing. Because Si and STO have different lattice parameters, the inconsistent referencing introduced a shift in the x-axis scaling, which made the FS-STO L-scan (Fig. S5c) appear slightly displaced relative to the STO substrate and as-grown films (Fig. S5a, b).

We apologize for the confusion. To ensure a consistent and physically meaningful comparison, we have now re-plotted the FS-STO L-scan in Fig. S5c using the STO(111) lattice spacing (see revised Fig. R2). The updated representation more accurately reflects the structural correspondence between the freestanding and as-grown STO films. We sincerely appreciate the reviewer for pointing out this important detail.

The revised Fig. S5 has also been updated in the supplementary information section.

Figure R2 shows the L-scan XRD patterns of (a) bare STO (111) substrate, (b) as-grown STO/LSMO/STO (111) heterostructure, and (c) freestanding STO (111) transferred onto a Si substrate referenced with respect to STO (111). The coordinates are normalized according to the lattice parameters of STO (111), where 1 reciprocal lattice unit (r.l.u.) is defined as $2\pi/a_{\text{STO}(111)}$.

Comments:

5. Have you measured the SHG response of WS₂ flakes on freestanding STO(111) membranes? Such data could further support the proposed symmetry breaking (polarization) effects at the STO surface and the associated (or enhanced) SHG signals.

Response:

We sincerely thank the reviewer for raising this important question. The request regarding the SHG response of WS₂ on freestanding STO(111) membranes is very relevant, particularly because the polar nature of the STO(111) surface can induce symmetry breaking and influence nonlinear optical signals. We would like to clarify that the freestanding STO membranes used in TEM observation experience an additional transfer and handling procedure before WS₂ is placed on them. This extra processing step inevitably introduces small distortions and local curvature, which makes it more difficult to maintain a well-defined twist angle over a large area. Since an accurate twist angle is essential for resolving moiré related SHG features, we performed all systematic SHG measurements on WS₂ placed directly on bulk STO(111) single-crystal substrates, where the stacking geometry can be defined more reliably.

In response to the reviewer's question, we have also carried out SHG measurements on WS₂ flakes placed on freestanding STO(111) membranes. As shown in Fig. R3, the freestanding WS₂/STO(111) samples also exhibit clear angle-dependent SHG modulation. The overall

behavior closely resembles the results obtained from WS₂ on STO(111) substrates. This indicates that the moiré-induced modulation and STO(111)-driven symmetry breaking remain effective even in the freestanding configuration and can be detected through the nonlinear optical response of WS₂. We truly appreciate the reviewer’s insightful suggestion, which motivated us to include these additional measurements.

Figure R3 SHG images of (a,b) WS₂/freestanding STO(111) along with the (c-f) polar plots measured at different twist angles. The scale bar is 60 μm.

Comments:

6. What is the rationale for choosing WS₂ in this study? How do other 2D-TMDs behave when interfaced with complex oxides? Please discuss this in the main text.

Response:

We thank the reviewer for this thoughtful question. We selected WS₂ as the model 2D-TMD in this study primarily because S-based TMDs are more stable and tolerant under ambient/room-temperature conditions, which allows us to carry out measurements with better reproducibility and reliability. In addition, monolayer WS₂ offers a large spin-orbit coupling that yields a well-defined excitonic structure, so interfacial potential modulations imposed by the complex oxide (e.g. polar surfaces, octahedral rotations, or long-wavelength structural modulations) can be clearly resolved through the A-exciton response. Finally, its hexagonal lattice provides a predictable lattice mismatch with perovskite oxides (here SrTiO₃), so for finite twist angles it naturally forms a moiré-type superlattice, enabling us to probe “oxide-imposed moiré” effects in a controlled manner.

Based on our structural simulations, a broad range of 2D materials can form a moiré-type superlattice on a perovskite oxide surface (see Fig. R4; SrTiO₃ is taken here as the

representative case). This is because the rotational symmetry of a hexagonal TMD, together with its lattice mismatch to a (111)-oriented perovskite surface, naturally generates a long-wavelength modulation at finite twist angles. In other words, the moiré geometry is not specific to the $\text{WS}_2/\text{SrTiO}_3$ combination.

Other TMDs (e.g., MoS_2 , MoSe_2 , WSe_2 , TaSe_2 , MoTe_2) are therefore expected to show similar moiré formation on oxide surfaces, but the actual lattice mismatch (see Table 1) and thus the moiré periodicity will differ, as illustrated in Fig. R4. For example, WSe_2 would produce a moiré pattern similar to WS_2 but with a slightly different period, whereas TaSe_2 and MoTe_2 , having larger mismatches, would form shorter-wavelength moiré patterns and may display weaker long-range ordering. For this reason, WS_2 offers the most balanced combination of lattice matching, optical visibility, and structural robustness to demonstrate the moiré mechanism in TMD/oxide heterostructures.

Consistently, in our experiments we have also identified that MoS_2 on SrTiO_3 (111) exhibits an angle-dependent SHG modulation, as shown in Fig. R5, which agrees with a moiré-induced modulation and effective symmetry lowering at the interface. These results support the more general conclusion that perovskite–TMD heterostructures can generically host moiré-like interfacial phenomena.

Once the geometric condition for forming a moiré superlattice is met, the key distinctions among other 2D TMDs arise from their intrinsic electronic structures (band gap and alignment, polarizability, metallic vs semiconducting nature) and the strength of their spin–orbit coupling. These material-specific factors dictate how strongly a given 2D layer can “sense” or respond to the slowly varying interfacial potential imposed by the complex oxide. Thus, even though MoS_2 , WSe_2 , MoSe_2 , MoTe_2 , or even metallic/CDW-type TMDs such as TaS_2 and TaSe_2 can all form moiré patterns with perovskite oxides in a similar geometric manner, their oxide driven moiré responses remain unexplored and may host novel interfacial phenomena beyond what we observe in WS_2 : As potential scenarios, semiconducting TMDs with weaker SOC will show a more modest moiré-induced band/valley reconstruction; Se- or Te-based TMDs with larger polarizability or stronger SOC can more efficiently imprint the oxide’s periodic potential into spin–orbit active moiré minibands; and metallic/CDW-prone TMDs are more likely to exhibit a moiré correlation or moiré CDW interplay rather than purely semiconducting moiré minibands. These scenarios, while speculative, suggest worthwhile directions for further exploration. In this sense, WS_2 serves as a representative platform for demonstrating oxide-imposed moiré modulation, while also pointing toward a broader class of TMD/oxide heterostructures that remain open for future exploration.

Our present WS_2 /oxide study demonstrates that complex oxides can effectively impose a moiré type modulation on 2D layers. Other 2D TMDs are expected to follow the same geometric route, but their distinct electronic structures will make different aspects of the moiré physics visible (spin–orbit, polarizability, or correlation dominated). Related experiments on these additional 2D/oxide combinations are underway, and once they are fully understood and explained, we will present them in separate publications in the near future. A concise and

focused discussion reflecting these points has been incorporated into the revised manuscript.

Table 1 Experimental in-plane lattice constants of selected monolayer transition-metal dichalcogenides (MX₂).

MX ₂	WS ₂	MoS ₂	WSe ₂	MoSe ₂	TaS ₂	TaSe ₂	MoTe ₂
a (Å)	3.153 ^{1,2}	3.16 ³	3.29 ⁴	3.293 ⁵	3.3 ⁶	3.43 ⁷	3.517 ⁵
Lattice Mismatch	1.1%	0.88%	3.1%	3.18%	3.54%	7.05%	9.35%

Figure R4 Simulated moiré superlattices of monolayer transition-metal dichalcogenides (MX₂) on SrTiO₃ substrates. (a–d) Top-view moiré patterns of WS₂, WSe₂, TaSe₂ and MoTe₂ monolayers on (111)-oriented SrTiO₃ with a 1° rotational mismatch. The periodicity of the moiré lattice decreases with increasing in-plane lattice constant of the overlying MX₂ layer. Scale bars are 5 nm.

Figure R5 SHG images of MoS₂/STO(111). The scale bar is 10 μm.

References

1. Schutte, W. J., De Boer, J. L. & Jellinek, F. Crystal structures of tungsten disulfide and diselenide. *J. Solid State Chem.* **70**, 207-209 (1987).
2. Mahatha, S. K. *et al.* Quasi-free-standing single-layer WS₂ achieved by intercalation. *Phys. Rev. Mater.* **2**, 124001 (2018).
3. Joensen, P., Crozier, E. D., Alberding, N. & Frindt, R. F. A study of single-layer and restacked MoS₂ by x-ray diffraction and x-ray absorption spectroscopy. *J. Phys. C: Solid State Phys.* **20**, 4043 (1987).
4. Zhang, Y. *et al.* Electronic structure, surface doping, and optical response in epitaxial WSe₂ thin films. *Nano Lett.* **16**, 2485-2491 (2016).
5. Caramazza, S., Marini, C., Simonelli, L., Dore, P. & Postorino, P. Temperature dependent EXAFS study on transition metal dichalcogenides MoX₂ (x = S, Se, Te). *J. Phys. Condens. Matter* **28**, 325401 (2016).
6. Silva, C. C. *et al.* Structure of monolayer 2H-TaS₂ on Au(111). *Phys. Rev. B* **104**, 205414 (2021).
7. Brown, B. E. & Beerntsen, D. J. Layer structure polytypism among niobium and tantalum selenides. *Acta Cryst.* **18**, 31-36 (1965).

Comments:

7. It would be beneficial to briefly mention potential applications to help attract a broader readership.

Response:

We thank the reviewer for this valuable suggestion. In the revised manuscript, we have now added a concise discussion highlighting the broader implications and potential applications of moiré-engineered TMD/oxide heterostructures.

Our results show that moiré formation at the WS₂/oxide interface provides a promising pathway toward hybrid semiconductor–oxide quantum materials. Such heterostructures offer opportunities for tunable optoelectronic, spintronic, and valleytronic functionalities, enabled by the strong dielectric response and diverse ferroic or correlated phases present in complex oxides. These functional orders, absent in conventional van der Waals stacks, allow external control over exciton energies, charge transfer, and interfacial coupling through electrostatic, strain, or phonon engineering. We now emphasize in the manuscript that moiré-engineered TMD/oxide systems may serve as a versatile platform for non-volatile optoelectronic,

spintronic, and quantum device concepts, where the oxide's ferroelectric, magnetic, or electronic order modulates the optical or excitonic states of the TMD layer.

In the revised manuscript, we have added the changes as below:

Looking ahead, our results establish a clear demonstration of moiré formation in WS₂/STO and WS₂/LSMO twisted heterostructures and position oxide–TMD interfaces as a versatile platform for exploring emergent quantum phenomena. The tunable moiré potential in these systems enables controlled modulation of exciton dynamics, charge transfer, and many-body interactions, where both interfacial dielectric screening and local strain fields arising naturally within the moiré supercell can act as complementary tuning knobs. Beyond WS₂, the framework we introduce can be broadly generalized to other TMDs with distinct excitonic and electronic structures, as well as to a wide family of functional oxides, including ferroelectrics, correlated nickelates, and magnetic manganites (Figure S21, & S29). Coupling excitonic TMD layers with these oxide order parameters opens the door to moiré-engineered ferroelectric modulation, magnetically tunable valley polarization, interfacial Mott transitions, and spin-textured moiré minibands. Such tunability is particularly appealing for designing valleytronic elements, excitonic modulators, and quantum photonic components. At the same time, the intrinsic electrostatic, mechanical, and ferroic tunability of oxide membranes provides dynamic control of twist-dependent coupling that is not accessible in conventional van der Waals moiré systems. Altogether, this work not only demonstrates the feasibility of creating robust oxide–TMD moiré superlattices but also highlights their unique potential as reconfigurable building blocks for low-power optoelectronic switches, integrated quantum photonic elements, and multifunctional hybrid quantum technologies fully compatible with existing oxide electronics.

Reviewer 4

Comments:

This manuscript studies the artificially engineered moiré superlattices formed by stacking and twisting a vdW-WS₂ on complex oxide materials (STO(111) and LSMO(111)). The authors observe the moiré exciton minibands, which confirms the emergence of moiré electronic structures. The authors also provide DFT calculations to support their findings. After getting through the previous reviewer's comments and the report, I am delighted to recommend this work for publication in Nature Communications. The reason is that the authors have supported their observations by theoretical input, which is convincing.

Response:

We sincerely thank the reviewer for the very positive and encouraging assessment of our work. It is truly our honor that the significance of our results has been recognized, and we are especially grateful for the reviewer's clear recommendation for acceptance. We deeply appreciate the supportive remarks and the confidence placed in our study, particularly the acknowledgement of the strength gained from the combined experimental and theoretical analyses. The thoughtful evaluations from all reviewers have greatly helped us refine the manuscript and further enhance its clarity and scientific rigor.

Reviewer 1&2

Comments:

I appreciate the response and additional experiments provided by the authors. Please be careful of the Raman notations - for TMD monolayers, the correct in-plane mode representation should be E' according to the monolayer symmetry. I believe this work meets the criteria for publication in Nature Communications.

Response:

We appreciate the reviewer's kind reminder. We agree that, according to the monolayer symmetry, the correct notation for the in-plane and out-of-plane Raman mode should be E' and A'1. We have now revised the Figs. S8 and S25 accordingly.

In this manuscript, the authors report the fabrication and characterization of twisted heterostructures composed of freestanding SrTiO₃ (111) thin membranes and monolayer WS₂. Using a combination of electron microscopy, optical spectroscopy (differential reflectance, PL and SHG), and ultrafast pump-probe measurements, the authors claim to observe moiré superlattice formation and associated quantum phenomena, including moiré exciton minibands, twist-angle dependent charge transfer, and nonlinear intensity modulation. This study aims to expand the landscape of moiré systems by integrating complex oxides -known for strong correlations and tunable order parameters- with van der Waals materials. The central finding -that moiré patterns form between monolayer WS₂ and STO (111)- is clearly supported by high-resolution TEM and diffraction data. The fabrication method involving freestanding epitaxial oxide membranes is technically solid and indeed offers a promising new route for interfacial engineering.

However, the manuscript suffers from a number of significant weaknesses that make me conclude it does not meet the standards of *Nature Communications*. While the experimental work is promising, particularly the demonstration of moiré pattern formation in hybrid oxide-TMD heterostructures, the interpretation of the data requires significant revision. The speculative claims need to be supported by theoretical calculations, and additional optical measurements (especially low-temperature PL) are needed to substantiate key conclusions.

1. The interpretation of spectroscopic data relies heavily on speculative claims of *quantum confinement, orbital hybridization, moiré exciton minibands, and band flattening*. No theoretical calculations or simulations are provided to support these claims. Without even basic electronic band structure modeling or a discussion of excitonic selection rules in this hybrid system, the interpretation goes well beyond what is warranted by the data. These parts of the manuscript should be either significantly revised or supported by theoretical input. For

instance, claiming exciton trapping within a moiré potential is highly speculative in the absence of any estimation of the potential depth and periodicity relative to the exciton Bohr radius.

2. Low-temperature PL spectra are entirely absent. These are essential to support the ΔR measurements and claims of discrete quantum states. Furthermore, PL (along with polarization-resolved PL and magneto-optics, as well as power-dependent and excitation energy-dependent PL experiments) could reveal signatures of exciton trapping and linewidth narrowing, helping to differentiate between moiré-induced effects and inhomogeneous broadening due to defects or strain.
3. The claim of novelty in forming moiré superlattices between oxides and TMDs is perhaps overstated. Prior reports have demonstrated similar systems. For example, *Mark J Haastrup et al 2023 J. Phys.: Condens. Matter 35 194001* discusses moiré formation at oxide-MoS₂ interfaces. The authors should cite this and clearly describe their technical and conceptual advances.
4. The reported PL spectra at room temperature show a dominant feature at ~ 1.97 eV, with no significant neutral exciton peak at ~ 2.00 eV, as typically observed in high-quality WS₂. This suggests different scenarios: (i) significant carrier doping in the WS₂, possibly due to intrinsic quality or (ii) charge-transfer effects from/to STO (thus, it is a trion). Another possibility would be (iii) screening of the Coulomb interactions from the high- κ dielectric environment (however the latter would result in a blueshift of the exciton, please see <https://pubs.acs.org/doi/full/10.1021/acsnano.4c11563>). The authors should clarify the origin of the main emission energy and discuss its impact on the observed optical features. Furthermore, the fitting presented in Figure S5 appears imprecise, with an unusually broad trion linewidth and a potentially incorrect assignment of the main peak to neutral excitons. A comparison with the reflectivity spectra would help clarify this issue. Notably, the room-

temperature excitonic emission is significantly broadened-comparable to, or even worse than, that observed in low-quality TMDs on SiO₂/Si substrates.

5. Regarding the observed variation in SHG intensity as a function of twist angle, it is unclear why simply the possibility of isotropic biaxial strain -potentially arising from symmetric moiré-induced buckling in the TMD layer- is not considered or discussed beyond complex charge transfer processes that sensitively depend on interlayer distance. This scenario could possibly contribute to the SHG modulation and should be addressed by the authors.
6. While the manuscript presents broad claims regarding the potential of oxide/TMD moiré systems for quantum materials and correlated phenomena, these statements remain overly general. A more concrete and focused discussion outlining specific applications and/or possible research directions-grounded in the demonstrated moiré formation in WS₂/STO heterostructures-is necessary to support the broader impact of this work.